# Analysis of stochastic fluctuations in responsiveness is a critical step toward personalized anesthesia

Andrew R McKinstry-Wu[1†], Andrzej Z Wasilczuk[1,2†], Benjamin A Harrison[1], Victoria M Bedell[1], Mathangi J Sridharan[3], Jayce J Breig[4], Michael Pack[5], Max B Kelz[1,2], Alexander Proekt[1*]

[1]Department of Anesthesiology and Critical Care, University of Pennsylvania, Philadelphia, United States; [2]Department of Bioengineering, University of Pennsylvania, Philadelphia, United States; [3]College of Medicine, The Ohio State University, Columbus, United States; [4]Department of Medicine, Drexel University College of Medicine, Philadelphia, United States; [5]Department of Medicine, University of Pennsylvania, Philadelphia, United States

**Abstract** Traditionally, drug dosing is based on a concentration-response relationship estimated in a population. Yet, in specific individuals, decisions based on the population-level effects frequently result in over or under-dosing. Here, we interrogate the relationship between population-based and individual-based responses to anesthetics in mice and zebrafish. The anesthetic state was assessed by quantifying responses to simple stimuli. Individual responses dynamically fluctuated at a fixed drug concentration. These fluctuations exhibited resistance to state transitions. Drug sensitivity varied dramatically across individuals in both species. The amount of noise driving transitions between states, in contrast, was highly conserved in vertebrates separated by 400 million years of evolution. Individual differences in anesthetic sensitivity and stochastic fluctuations in responsiveness complicate the ability to appropriately dose anesthetics to each individual. Identifying the biological substrate of noise, however, may spur novel therapies, assure consistent drug responses, and encourage the shift from population-based to personalized medicine.

**\*For correspondence:**
proekt@gmail.com

[†]These authors contributed equally to this work

**Competing interests:** The authors declare that no competing interests exist.

## Introduction

One of the great promises of personalized medicine is the delivery of a maximally efficacious and minimally harmful dose of appropriate medication to every patient (*Fitzgerald et al., 2006*). Traditionally, dosing decisions are based on the relationship between drug concentration and the magnitude of effect observed in a population expressed as the sigmoidal dose-response curve (*Goodman, 1996*). An implicit assumption of this approach is that population averages adequately reflect the processes operating within each individual patient. This is not always true. For many drug classes such as antiepileptics, anesthetics, and antiarrhythmics, the response is binary at the level of an individual—the desired effect is either present or not (*Löscher, 2011*; *Sunderam et al., 2001*; *Jürgens et al., 2003*; *Sonner, 2002*). The population-based dose-response curve, in contrast, is a smooth graded function of drug concentration. Therefore, in order to deliver on the promise of optimal drug dosing at the individual level, the relationship between individual binary responses and the population-based graded estimates of drug potency have to be more rigorously defined (*Koch-Weser, 1975*).

Many examples of graded population-level responses co-existing with discrete, binary individual responses are seen in biophysics. A priori, one might have expected that the current flow through a

**eLife digest** Every year, millions of patients undergo general anesthesia for complex or life-saving surgeries. In the vast majority of cases, the drugs work as intended. But a minority of patients take longer than expected to regain consciousness after anesthetic, and a few wake up during the surgery itself. It is unclear what causes these unintended events.

When choosing an anesthetic dose for each patient, physicians rely on data from large clinical studies. These studies expose many patients to different doses of an anesthetic drug. At higher doses, fewer and fewer patients remain conscious. This enables physicians to identify the dose at which an average person will lose consciousness. But this approach ignores the difference between the response of an individual and that of the population as a whole. At the population level, the likelihood of a patient being awake decreases smoothly as the concentration of anesthetic increases. But within that population, each individual patient can only ever show a binary response: awake or not awake.

To compare anesthetic effects on individuals versus populations, McKinstry-Wu, Wasilczuk et al. exposed mice to a commonly used anesthetic called isoflurane. During prolonged exposure to a constant dose of the drug, each mouse was sometimes unconscious and sometimes awake. These fluctuations in responsiveness seemed to occur at random. Exposing zebrafish to propofol, an anesthetic that works via a different mechanism, had a similar effect.

Notably, the responses of both species to anesthesia showed a phenomenon known as inertia. If an individual was unresponsive at one point in time, they were likely to still be unresponsive when assessed again after three minutes. The amount of inertia was similar in mice and zebrafish. This suggests that the mechanism responsible for inertia has remained unchanged over more than 400 million years of evolution.

The results reveal similarities between how individuals respond to anesthetics and how individual anesthetic molecules act on cells. When a molecule binds to its receptor protein on a cell, the receptor fluctuates spontaneously between active and inactive states. Studying how individuals respond to drugs could thus provide clues to how the drugs themselves work. Future studies should explore the biological basis of fluctuations in anesthetic responses. Understanding how these arise will help us tailor anesthetics to individual patients.

single ion channel molecule would simply be a scaled version of the current recorded in a cell containing many such ion channels. Yet, this is not the case (*Sakmann and Neher, 1984*). While at the whole cell level, current varies smoothly as a function of voltage and time (*Hodgkin and Huxley, 1952*), each ion channel molecule stochastically switches between conductive and nonconductive states (*Fukushima, 1982*; *Hoshi et al., 1990*; *Aldrich et al., 1983*). Ligand-gated ion channels also stochastically fluctuate between discrete conductive and non-conductive states at a constant concentration of a ligand (*Brickley et al., 1999*; *Swanson et al., 1996*; *Papke et al., 1989*). Stochastic transitions between discrete states extend to higher levels of organization. For example, synaptic transmission is mediated by stochastic vesicular fusion events (*del Castillo and Katz, 1954*; *Kuffler and Yoshikami, 1975*; *Rizzoli and Betz, 2005*). At the cellular level, stochastic fluctuations in gene expression and protein abundance are observed even in a clonal population of cells (*Elowitz et al., 2002*; *McAdams and Arkin, 1997*; *Newman et al., 2006*; *Raser and O'Shea, 2004*). For instance, stochastic forces influence differentiation patterns across identical cells and give rise to drastically different proteomic responses of clonal cancer cells to drugs (*Cohen et al., 2008*; *Losick and Desplan, 2008*). Thus, stochastic fluctuations among discrete states even in a constant environment are the rule at the microscopic level in many, if not all, biological processes.

Detailed quantitative models of stochastic switching between different states of an individual molecule (*Hille, 2001*; *Colquhoun and Hawkes, 1995*) are required to forge the relationship between discrete microscopic events and graded responses in the population. Macroscopic constructs such as pharmacologic efficacy depend upon drug-induced changes in discrete states of single molecules as well as stochastic switches among them. Yet, drug efficacy does not uniquely specify responses of individual molecules (*Colquhoun, 1998*). Thus, while discrete responses at the

individual level can be linked to the smooth population-level responses using techniques from statistical mechanics, individual responses cannot be readily inferred from population-level analyses.

Here, we asked whether stochastic fluctuations among discrete states, similar to those postulated to act at the level of individual molecules, organelles, and cells influence the responses of intact multicellular organisms exposed to a constant drug concentration. For this purpose, we chose to focus on anesthetics. Anesthesia is thought to consist of four basic components: amnesia, immobility, analgesia, and unconsciousness. Responses to stimuli are typically used to quantify the various components of anesthesia. At lower anesthetic doses that produce light sedation, a patient can respond to salient verbal commands and less noticeable auditory commands (*Wong et al., 2014*), albeit with longer latency. When the state of general anesthesia is attained, the patient is unresponsive even to a painful surgical stimulus (*Miller, 2014*). Clinical assessment of anesthetic depth ultimately collapses to binary outcomes: amnestic or not, immobile or not, and conscious or not. We consequently studied binary responses to a simple stimulus at a fixed anesthetic concentration across individuals to determine whether they are anesthetized or not. These all-or-none responses can be assessed by determining whether a human patient reacts to a simple verbal command (*Sanders et al., 2017*; *Russell, 2013*; *Flaishon et al., 1997*). The concentration of anesthetic at which 50% of patient lose their ability to respond to verbal commands is known at MAC-awake (*Franks, 2008*). In rodent literature, the righting reflex (RR) is typically used to establish whether a mouse is awake or not (*Figure 1A*) (*Wasilczuk et al., 2018*; *Franks, 2006*). MAC-awake and the $EC_{50}$ for loss of RR are closely correlated across mechanistically distinct anesthetics (*Franks, 2008*). Discrete responses to simple stimuli can be used in other organisms such as larval zebrafish to assess anesthetic potency (*Yang et al., 2018*). In contrast to discrete measures used in individual subjects, anesthetic potency at a population level is expressed as a smooth sigmoid function (*Miller, 2014*). Traditionally, the dichotomy between the graded population-level response on the one hand, and the binary responses of individuals on the other, has been interpreted as inter-subject variability (*Sonner et al., 2000*). For instance, at the population level half maximal effective concentration ($EC_{50}$), it is assumed that half of all the subjects will be able to consistently respond to the stimulus while the other half will consistently fail to do so (*Figure 1B*). Yet, this is not the only possibility. It is possible that at $EC_{50}$, each individual subject will respond to 50% of stimuli (*Figure 1C*). Finally, it is possible that the probability of response is influenced by the state of the subject at the moment when the stimulus is applied (*Figure 1D*).

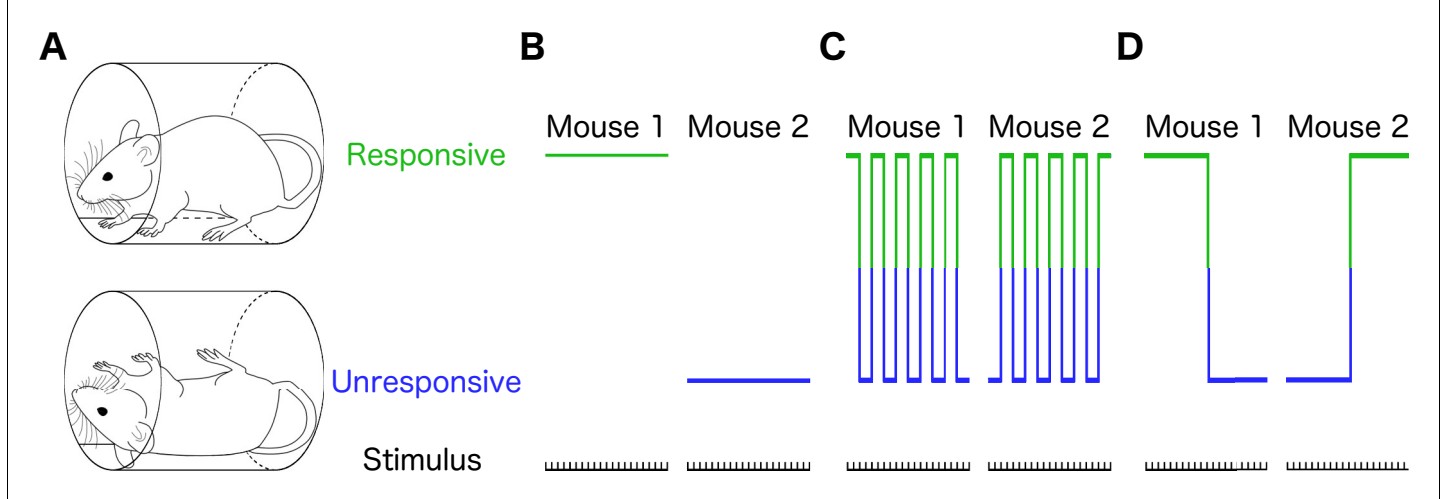

**Figure 1.** Different individual behaviors can produce identical population effects. (**A**) Presence (top) or absence (bottom) of righting reflex is a binary behavioral measure that distinguishes awake from anesthetized mice. Schematic representations of three qualitatively distinct models that yield 50% population responsive probability (PRP) are shown in B-D. (**B**) Responses in each individual remain constant on repeated righting reflex assessments. Hence, PRP is solely a consequence of individual differences in drug sensitivity. This is the commonly assumed model. (**C**) PRP is a consequence of a constant response probability for each animal on each trial; responses on repeated trials fluctuate, but the probability of response is independent of the state of the animal. (**D**) PRP is a consequence of a state-dependent fluctuations in responses to repeated presentation of the stimulus. In this example, animals tend to stay in the previously observed state.

To distinguish among these possibilities, we exposed both mice and zebrafish to fixed concentrations of two mechanistically distinct anesthetics and repeatedly tested their responsiveness. The results unequivocally demonstrate that at a fixed drug concentration, each individual mouse or zebrafish stochastically switches between being responsive and unresponsive. The probability of response depends upon the subject's state. Therefore, stochastic state switching is apparently present at all organizational levels, from receptor to cell to whole animal. Unlike identical ion channel molecules, however, individual animals both in highly genetically inbred and across outbred populations exhibited dramatic variability in parameters that describe the stochastic switching between the responsive and unresponsive states. The inter-individual variability in parameters of the stochastic model fit to each individual was highly structured in both mice and in zebrafish. One manifestation of this structure is that while the overall sensitivity to anesthetics varied among individuals, the amount of noise that drives state switches was the same across individuals exposed to the same anesthetic concentration.

As a result of the inter-subject variability and stochastic intra-subject fluctuations in responsiveness, the population-level concentration response cannot be used to reliably determine the probability that any given individual will (or will not) be anesthetized. Detailed quantification of stochastic forces that shape the within-subject fluctuations in responsiveness and between-subject variability lays the foundation for the construction of more informative stochastic models that can reconcile binary responses of individual organisms with population-based measures of drug potency. These models can in turn be used to deliver on the promise of personalized medicine to deliver the appropriate dose of medication tailored to each individual patient.

## Results

### Individual responses to anesthetics fluctuate in a state-dependent fashion at a fixed drug concentration

The presence of the righting reflex (RR) during continuous isoflurane administration was used as the measure of responsiveness in mice. After 2 hr of equilibration at 0.6% isoflurane, the probability of an intact RR was 44 ± 6% (mean ± SD across trials). This did not correspond to 44% of mice being consistently responsive and the rest being consistently unresponsive, as is commonly assumed. Instead, in every mouse, the outcome of the RR test fluctuated over time at a fixed anesthetic concentration (*Figure 2A*) while the population response probability at 0.6% isoflurane remained stable over time (*Figure 2B*). Individual fluctuations in RR at concentrations deviating from $EC_{50}$ were less frequent, but nevertheless were reliably observed at concentrations below 0.9% (*Figure 2—figure supplement 1*).

In an analogous experiment using a mechanistically distinct anesthetic, propofol, we determined the responsiveness in larval zebrafish using the startle reflex (SR) triggered by mechanical stimulation. Individual larval zebrafish demonstrated fluctuating responses to identical tap stimuli (*Figure 2C*). Zebrafish exposed to no propofol had a significantly higher response probability to the tap stimulus (*Figure 2—figure supplement 2A*, $U = 55$, $n_{E3} = n_{3\mu M}=360$ trials, p<*0.0001*). Drug concentration remained constant in the propofol exposures (*Figure 2—figure supplement 2B*). The response probability across the population remained constant (45 ± 5% mean ± SD across trials) for three hours after an initial hour of equilibration in 3 µM propofol (*Figure 2D*).

Effective concentration, defined as the concentration of the drug required to produce an effect of a given intensity, is a universally used population-based measure of drug potency (*Goodman, 1996*). We sought to apply this measure to each individual. At the individual level, effective concentration is equivalent to the average across-trial probability of observing a response to a stimulus at a given drug concentration. Therefore, we compared the experimental results in *Figure 2A and C* to simulations of a Bernoulli process (Materials and methods) constructed such that the probability of positive RR (or SR) was identical on each trial and the same as that observed experimentally for a population (*Figure 3A*). In mice for instance, this simulation was constructed such that the probability of responding to a stimulus is 44% on every trial. As expected, both simulations (*Figure 3A*) and the experimental results (*Figure 3B*) have a similar overall response probability (45 ± 3% and 44 ± 6% mean ± SD for simulations and mouse experiments, respectively). The median response probability in simulations was not statistically different from experimental observations in both mice (*Figure 3C*,

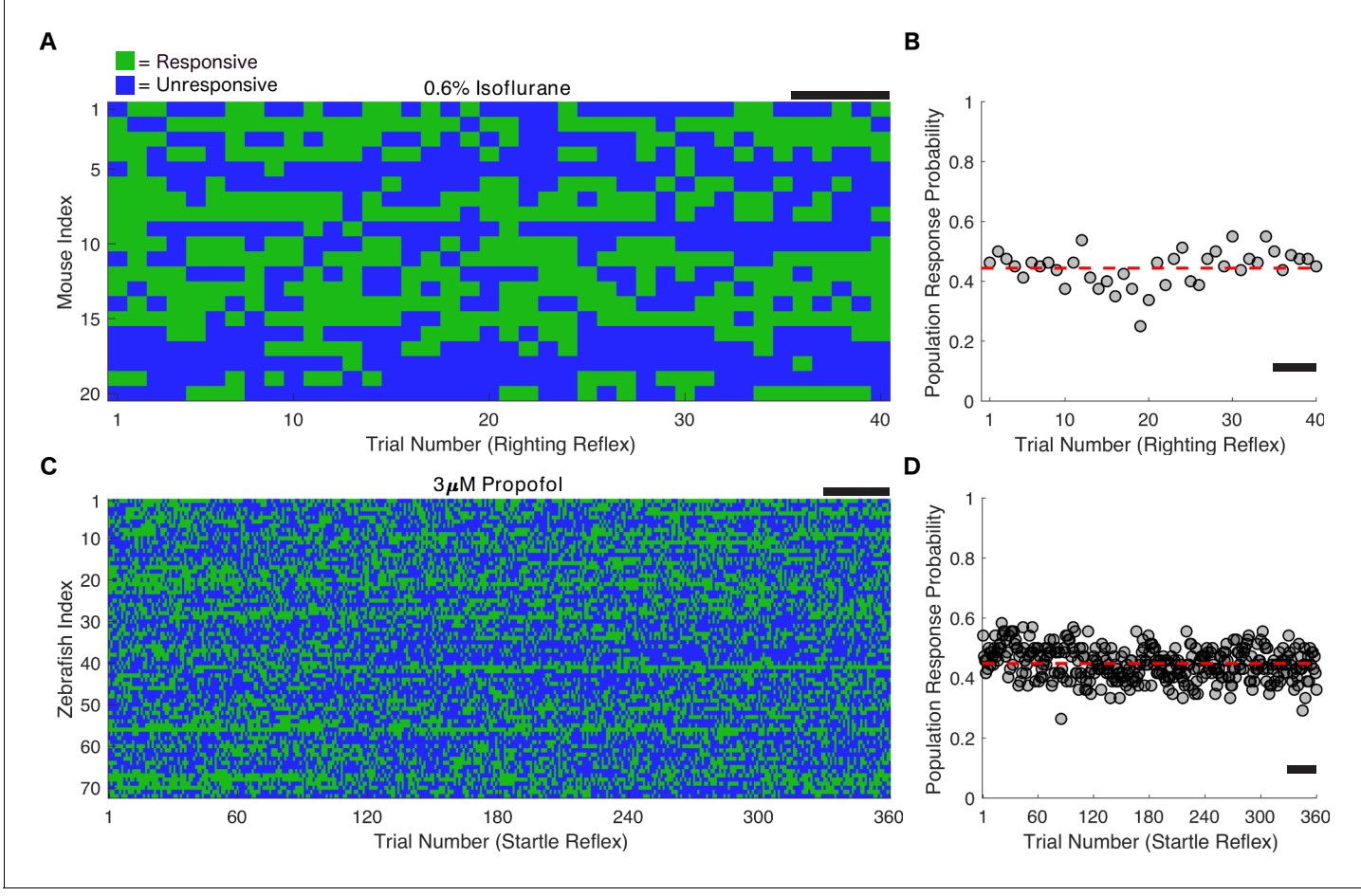

**Figure 2.** Individual responsiveness fluctuates over time with unchanging population response. (A) Twenty mice were exposed to 0.6% isoflurane for 2 hr, then righting reflex was assessed every 3 min for another 2 hr at the same isoflurane concentration. Fluctuations between responsive (green) and unresponsive (blue) states on repeated trials are seen in every animal. (B) Population response probability (PRP) remains constant over the duration of the entire experiment ($R^2 = 4.3 \times 10^{-16}$, p=1 correlation between trial number and response probability averaged across animals for each trial, Pearson's R). PRP averaged across animals and trials at 0.60% isoflurane is 0.44 ± 0.06 (mean ± SD, red line). (C) 72 larval zebrafish (five dpf) were exposed to 3 µM propofol for a total of 4 hr, with startle reflex assessed every 30 s for the final 3 hr of the exposure. Fluctuations between responsive and unresponsive states on repeated trials are also seen in every animal. (D) PRP remains constant over the duration of the entire experiment ($R^2 = 1.2 \times 10^{-14}$, p=1, correlation between trial number and response probability averaged across animals for each trial, Pearson's R). PRP averaged across animals and trials at 3 µM propofol is 0.45 ± 0.05 (mean ± SD, red line). Scale bars represent 15 min. Source data for population response probabilities used for this analysis are available in the *Figure 2—source data 1*.

The online version of this article includes the following source data and figure supplement(s) for figure 2:

**Source data 1.** Mean population response probabilities of righting reflex assays in mice at 0.6% isoflurane and startle reflex assays in zebrafish at 3µM propofol.

**Figure supplement 1.** Response probabilities in mice at varying isoflurane concentrations.

**Figure supplement 2.** Characterization of zebrafish startle response.

**Figure supplement 3.** Chamber equilibration occurs within 5 min.

---

$U = 197.5$, $n_{sim} = n_{exp} = 20$, p=0.95) and in zebrafish (*Figure 3D*, $U = 2276$, $n_{sim} = n_{exp} = 72$, p=0.34). Note, however, that the experimentally observed switches between positive and negative RR occur less frequently than in the simulation (*Figure 3E*, $U = 48.5$, $n_{sim} = n_{exp} = 20$, p<0.0001). This was also true of zebrafish (*Figure 3F*, $U = 300.5$, $n_{sim} = n_{exp} = 72$ p<0.0001).

To further quantify this resistance to state transitions, we compared the probability of becoming unresponsive after responding to a stimulus on a previous trial, $P(U|R)$ (*Figure 3B*, red arrow), to the probability of failing to respond on two consecutive trials, $P(U|U)$ (*Figure 3B*, purple arrow). In both mice (*Figure 3G*, $U = 103$, $n_{P(U|R)} = n_{P(U|U)}=20$, p<0.001) and zebrafish (*Figure 3H*, $U = 501$, $n_{P(U|R)} =$

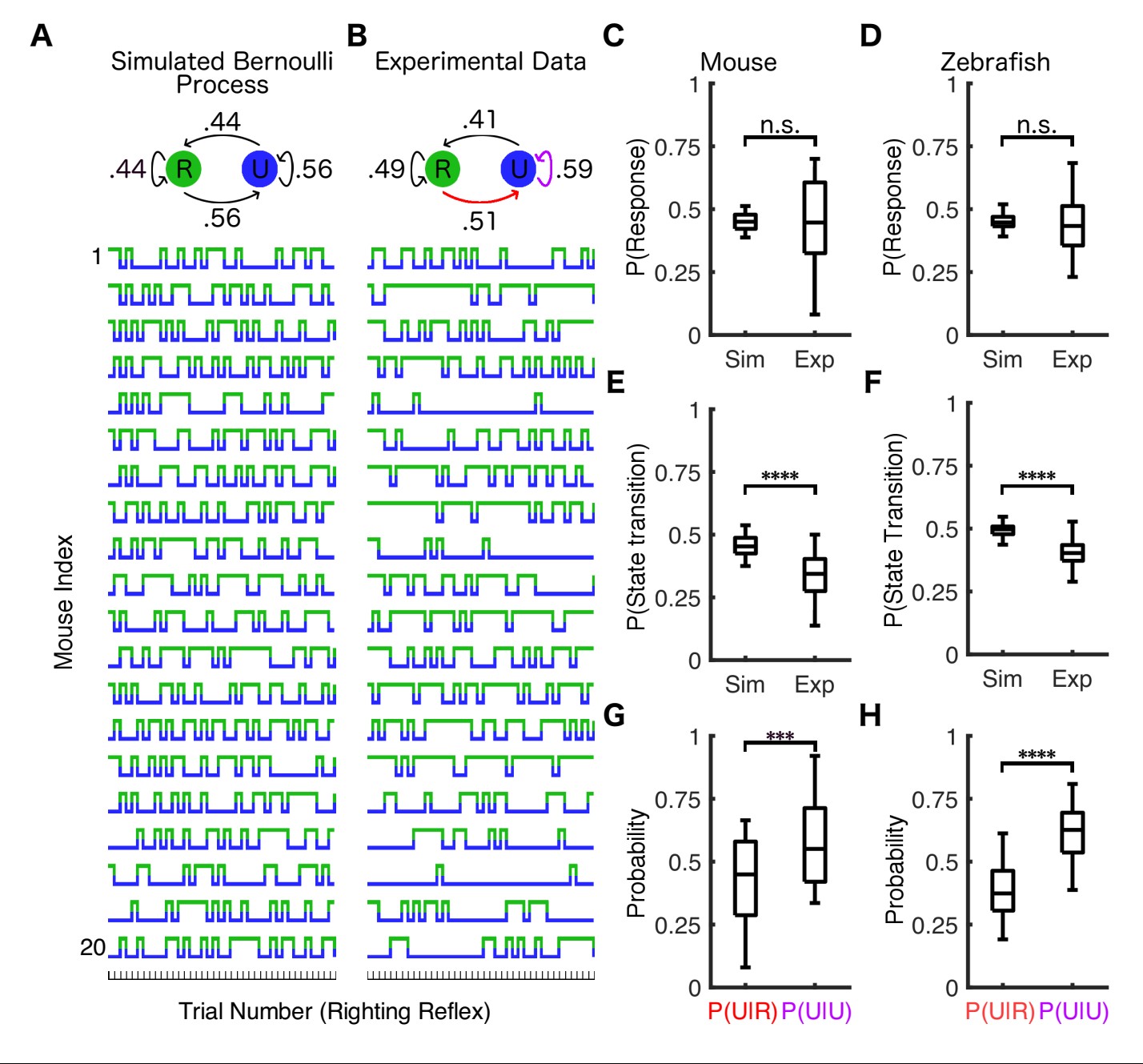

**Figure 3.** Response probability is state-dependent. (**A**) Twenty simulations of a Bernoulli process where the response probability is 44% for all trials (identical to the average PRP in mice at 0.6% isoflurane) were performed. Each simulation consisted of the same number of trials as the experimental data from (**B**). Similar simulations were conducted for zebrafish using PRP of 0.45. Simulated (**A**) and experimentally observed (**B**) transition probability matrices are shown schematically above traces. The population response probability in simulations is not statistically different from experimental observations across mice (**C**) ($U = 197.5$, $n_{sim} = n_{exp} = 20$, p=0.95) or across zebrafish (**D**) ($U = 2276$, $n_{sim} = n_{exp} = 72$, p=0.34). Both mice (**E**) ($U = 48.5$, $n_{sim} = n_{exp} = 20$, p<0.0001) and zebrafish (**F**) ($U = 300.5$, $n_{sim} = n_{exp} = 72$ p<0.0001) have fewer state transitions than simulations of the Bernoulli process. Both mice (**G**) ($U = 103$, $n_{P(U|R)} = n_{P(U|U)}=20$, p<0.001) and zebrafish (**H**) ($U = 501$, $n_{P(U|R)} = n_{P(U|U)}=72$, p<0.0001) are more likely to stay unresponsive if they failed to respond in the previous trial (purple arrow in **B**) than if they were able to respond to the previous stimulus (red arrow in **B**). In plots (**C–H**) box plots show inter-individual differences in estimated parameters (central mark indicates the median, box shows interquartile range, whiskers extend to the most extreme data points). Statistical significance is shown by ***p<0.001 ****p<0.0001. Source data for all quantitative analyses described are available in the *Figure 3—source data 1*.

The online version of this article includes the following source data for figure 3:

**Source data 1.** Individual simulated and experimental response probabilities and probability of transitions (total, P(U|U), and P(U|R)).

$n_{P(U|U)}$=72, p<*0.0001*), the probability of being unresponsive on the next trial was significantly higher if the animal was found to be unresponsive on the preceding trial. Hence, fluctuations in responsiveness under constant anesthetic concentration are inconsistent with a Bernoulli process. Therefore, while effective concentration is a useful measure of population-based drug potency, it cannot be adequately applied to an individual. Specifically, effective concentration cannot account for the apparent resistance to state transitions.

## Individual differences are highly structured

We now turn to the relationship between trial-to-trial fluctuations and population level variability. One common assumption is that observing a single mouse over long period of time is equivalent to observing a snapshot of the population of mice. In other words, an experiment on each individual mouse can be thought of as a different realization of the same process. In an apparent departure from this assumption, we observed high inter-individual variability in responsiveness in mice and in zebrafish. For instance, at 0.6% isoflurane some mice were able to right themselves on fewer than 20% of trials, while other mice within the same highly inbred population, exposed to the same anesthetic concentration, during the same experiment, were able to right themselves on ~70% of trials (*Figure 3C*). Experimentally observed inter-individual variability was significantly higher than in simulations constrained to have the same number of trials (*Figure 3C*, F(1, 38)=27.8, p<*0.0001* for mice, *Figure 3D*, F(1,142)=52.5, p<*0.0001* for zebrafish, Brown-Forsythe test). This implies that the observed inter-subject variability in responsiveness is unlikely to be solely due to finite sample size. Rather, anesthetic sensitivity can differ significantly between individuals.

We then sought to determine how inter-individual variability is reflected in the parameters of a model of trial-to-trial fluctuations in responsiveness fit to each animal individually (Materials and methods). The dwell times in responsive and unresponsive states for both mice (*Figure 4—figure supplement 1A*) and zebrafish (*Figure 4—figure supplement 1B*) were approximately exponential. No significant autocorrelations in fluctuations were observed (*Figure 4—figure supplement 1C,D*). Thus, switching between states of responsiveness and unresponsiveness in each individual can be well approximated by a two-state transition probability matrix (Materials and methods). Because the sum of transition probabilities in each row of this matrix is exactly one, the two-state transition probability matrix is completely specified by knowing the two transition probabilities along the diagonal, *P(U|U)* and *P(R|R)*. The plane spanned by these two diagonal transition probabilities, therefore, is the parameter space for models of stochastic fluctuations in responsiveness in mice and in zebrafish (*Figure 4A*). Movement along the x-axis to the right, results in decrease in the overall probability of response (e.g. from I to II and from III to IV, *Figure 4A*). Conversely, movement up along the y-axis results in the increase in the overall probability of response (e.g. from III to I and from IV to II, *Figure 4A*). Moving along the dotted lines within the parameter space does not affect the overall probability of being responsive – that is systems II and III have the same overall probability of response. The key difference between II and III is the number of transitions between the responsive and the unresponsive state. In contrast, movement from I to IV does not affect the sum of *P(U|U)* and *P(R|R)*. As a result, while the overall probability of responsiveness in I and IV is different, the amount of noise that drives state transitions between the responsive and the unresponsive states is the same. To realize why this is the case, we can visualize the two-by-two transition probability matrix as an energy landscape with two wells which correspond to the responsive and the unresponsive states (*Proekt and Hudson, 2018*). Conservation of the sum of *P(U|U)* and *P(R|R)* reflects the fact that the sum of the depths of the wells is conserved.

To characterize inter-individual variability, we estimated transition probability matrices for each animal individually. This yields a single estimate of P(U|U) and P(R|R) for each individual. The simplest model of inter-individual variability is that the transition probability matrix, *M*, for each individual is a random sample taken independently from the distribution of *P(U|U)* and *P(R|R)*. This null hypothesis corresponds to a cloud of points in *Figure 4B*. Yet, in stark departure from this prediction, the observed joint distribution of transition probabilities in individual mice lies on a diagonal (*Figure 4C*). Therefore, *P(U|U)* and *P(R|R)* are strongly negatively correlated among individuals ($R^2 = -0.85$, p<*0.0001*, Pearson's R). The same relationship was observed in larval zebrafish (*Figure 4E*, $R^2 = -0.77$, p<*0.0001*, Pearson's R). A similar strong negative correlation is observed when comparing decay rates of the responsive and unresponsive states estimated from individual dwell time

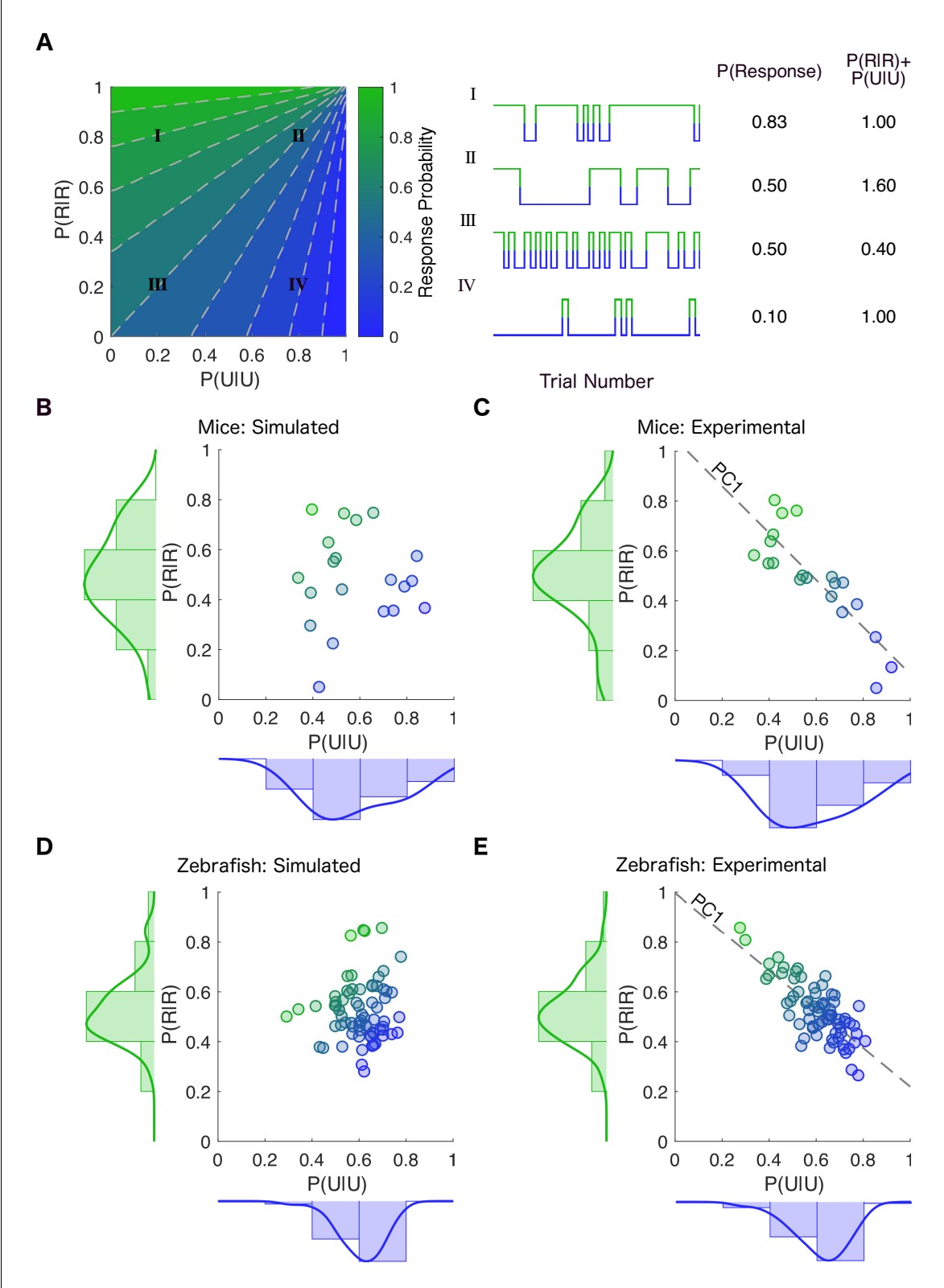

**Figure 4.** Stability of responsiveness and unresponsiveness are linked. (A) (Left) The transition probability matrix for each individual can be represented as a point on a plane spanned by the diagonal elements P(U|U) and P(R|R) denoting the probability of staying in the previously observed state. The plane is colored according to the across-trial response probability given by the transition probability matrix at each location in the plane. (Center) Examples of simulations constructed by creating transition probability matrices from the corresponding locations in the parameter plane. (Right)

*Figure 4 continued on next page*

*Figure 4 continued*

Movement along the dashed gray lines preserves the overall probability of response (Traces II and III). As one moves along the dashed gray line from bottom left to top right, the frequency of state transitions decreases but the overall response probability stays the same. In contrast, movement parallel to the line y = −x results in changes in response probability while conserving the noise driving transition between the responsive and the unresponsive states (Traces I and IV). This is because the line y = −x preserves the sum of stability of the responsive and the unresponsive states. (B) Transition probability matrices were empirically estimated for each mouse at 0.6% isoflurane. The empirically derived distributions of P(U|U) and P(R|R) (shown on the margins) were independently randomly sampled to create simulated individual points in the plane spanned by P(U|U) and P(R|R). Each point's color shows the average across- trial response probability estimated empirically using simulations constrained to have the same number of trials as experimental observations. (C) Distribution of transition probabilities experimentally observed in individual mice at 0.6% isoflurane. Inter- individual variability is constrained such that the sum of P(U|U) and P(R|R) is approximately the same ($R^2$ = -0.85, p < 0.0001, Pearson's R ) in individual mice. (D) A similar estimation and simulation was computed for zebrafish with P(U|U) and P(R|R) assumed to be independent of each other. (E) Distribution of the experimentally observed transition probabilities for individual zebrafish ($R^2$ = -0.77, p < 0.0001, Pearson's R ). While each individual transition probability matrix is defined by two parameters, the inter-individual variability in these matrices is well approximated (~90% of variance in both mice and zebrafish) by a single value corresponding to the projection onto the first principal component (PC1) shown by the dotted line in (C) and (E). Source data for experimental plots described are available in the *Figure 4—source data 1*.

The online version of this article includes the following source data and figure supplement(s) for figure 4:

**Source data 1.** Individual mouse and zebrafish P(U|U) and P(R|R) values.
**Figure supplement 1.** Estimates for dwell time distributions and autocorrelation functions from individual behavioral assays.

---

distributions fit to single exponential decay functions (dwell times from *Figure 4—figure supplement 1A,B*, $R^2_{mouse} = −0.64$, p=0.002, Pearson's R, $R^2_{zebrafish} = −0.71$, p<0.0001, Pearson's R).

Note that *P(U|U)* and *P(R|R)* quantify the propensity of the system to stay in its previously observed state. The fact that their sum is constant across individuals implies that the amount of noise that drives state transitions between responsive and unresponsive states was consistent in all individuals exposed to a fixed anesthetic concentration. Yet, sensitivity to anesthetics measured as the overall probability of responding to a stimulus varied broadly in the same population of individuals. Altogether these results indicate that, while transitions between states of responsiveness are noise-driven and therefore unpredictable, the amount of noise is tightly controlled in all individuals. Too little noise would result in individuals being trapped in a single state, whereas too much noise would overpower the intrinsic dynamics of the brain, leading to a noise dominated process characterized by rapid state switching.

## Individual differences in drug sensitivity complicate decoding of drug concentration from drug responses

Sigmoid dose-response curves are one-to-one functions—knowing drug concentration is sufficient to estimate probability of a response in a population. Critically, the converse is also true; knowing the probability of a response in a population is sufficient to determine the concentration of the drug to which this population is exposed. While this one-to-one relationship may hold for a population, we sought to determine whether the large inter-individual variability (*Figures 4–5*) complicates this one-to-one relationship at the level of an individual. We compared individual transition probability matrices estimated at 0.6% and 0.3% isoflurane. Note that here, mice exposed to 0.6% isoflurane were distinct from mice exposed to 0.3% isoflurane. Individual matrices estimated for 0.3% isoflurane, much like those estimated at 0.6% isoflurane, exhibited strong negative correlations ($R^2 = −0.86$, p<0.0001, Pearson's R) between the diagonal elements of the transition probability matrices (*Figure 6A*). At the level of the population, there were statistically significant differences between righting probability observed at the two anesthetic concentrations (*Figure 6B*, $U = 54$, $n_{0.3\% \ iso} = 18$, $n_{0.6\% \ iso} = 20$, p<0.001). Note, however, that there is significant overlap (57%) between the two distributions. As a consequence of this overlap, it is not possible to reliably determine whether an individual was exposed to 0.6% or 0.3% isoflurane over a large range of individual anesthetic sensitivities (*Figure 6C*). The same phenomenon was observed for individual zebrafish exposed to medium alone or 3 μM propofol (*Figure 6—figure supplement 1*). This observation is in stark contrast to the population-based measures of anesthetic potency. Hill slopes of the population-based dose-response curves for anesthetics are some of the steepest among clinically useful drugs (*Friedman et al., 2010*; *Joiner et al., 2013*). Thus, small increases in drug concentration are expected to result in a dramatic change in the probability that an individual will be anesthetized. Yet, as demonstrated here, there is considerable overlap

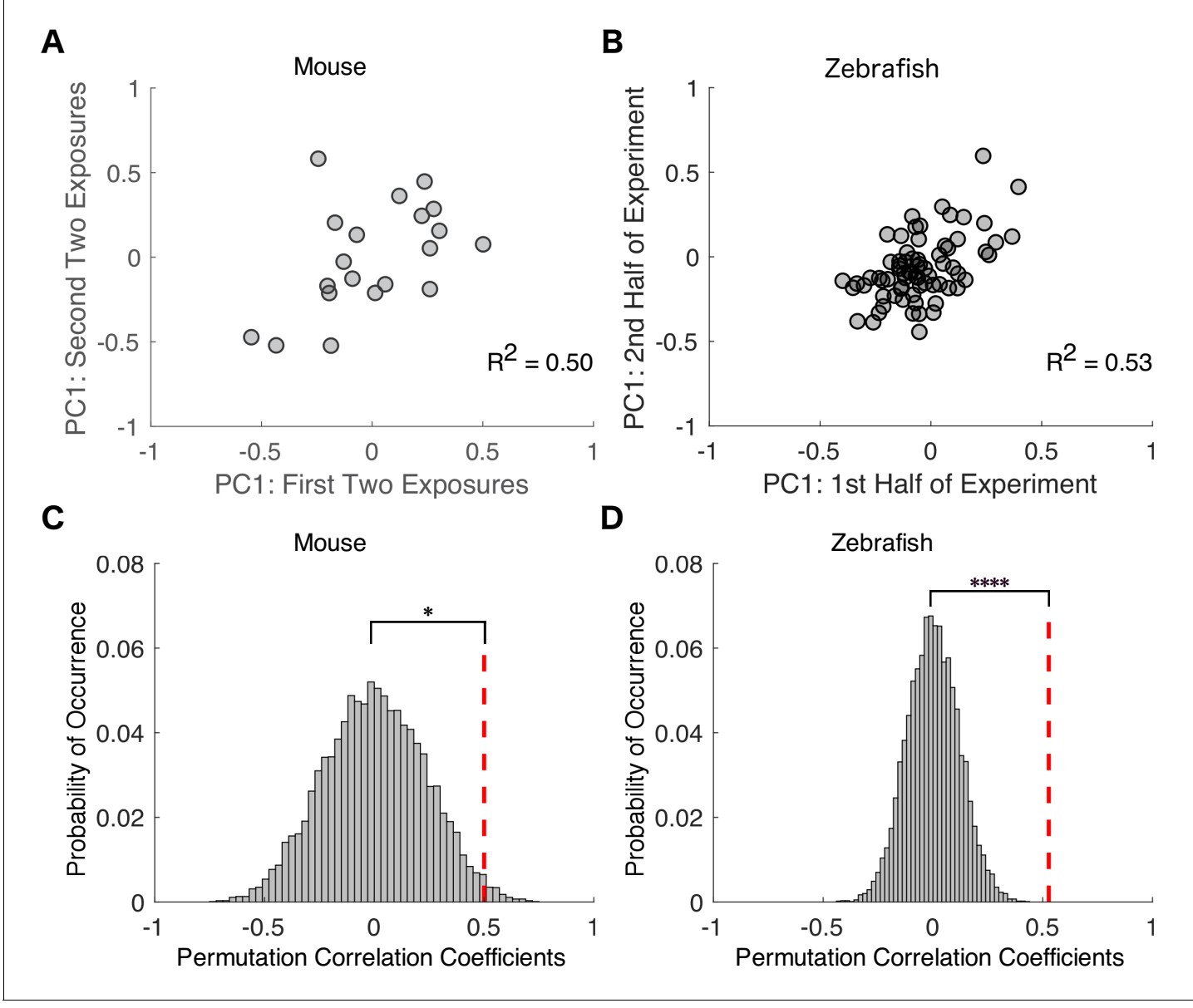

**Figure 5.** Variability in drug sensitivity is a stable characteristic of an individual at steady state equilibrium. Transition probability matrices were independently estimated for each mouse across the first two and last two of exposures to isoflurane and projected onto the first principal component. For zebrafish, transition probability matrices were estimated for the first and second halves of propofol exposure. Negative values represent animals that had a high probability of staying responsive and low probability of staying unresponsive, whereas positive values represent animals that had a high probability of staying unresponsive and low probability of staying responsive. (A) In mice, individual transition probability matrices estimated separately for the first two and the last two exposures to 0.6% isoflurane were correlated ($R^2 = 0.50$, p=0.025, Pearson's R ). (B) In zebrafish, individual transition probability matrices estimated separately for the first and second halves of the exposure to 3 µM propofol were correlated ($R^2 = 0.53$, p<0.0001, Pearson's R ). (C) Permutation test reveals that, in mice, experimentally observed within-subject correlations are significantly higher than in inter-subject shuffled surrogates (p=0.013). (D) Permutation test computed for zebrafish (p<0.0001). Statistical significance is shown by *p<0.05 ****p<0.0001. Source data for correlation plots described are available in the *Figure 5—source data 1*.

The online version of this article includes the following source data for figure 5:

**Source data 1.** PC1 values in mice for exposures 1/2 and 3/4 and in zebrafish for first and second half of exposure.

in sensitivity to anesthetics. This illustrates the fundamental inconsistency between population-based and individual-based measures of drug potency.

Because results in *Figure 6* were obtained in two separate cohorts of mice each exposed to a single anesthetic concentration, we sought to determine whether it is possible to reliably infer drug

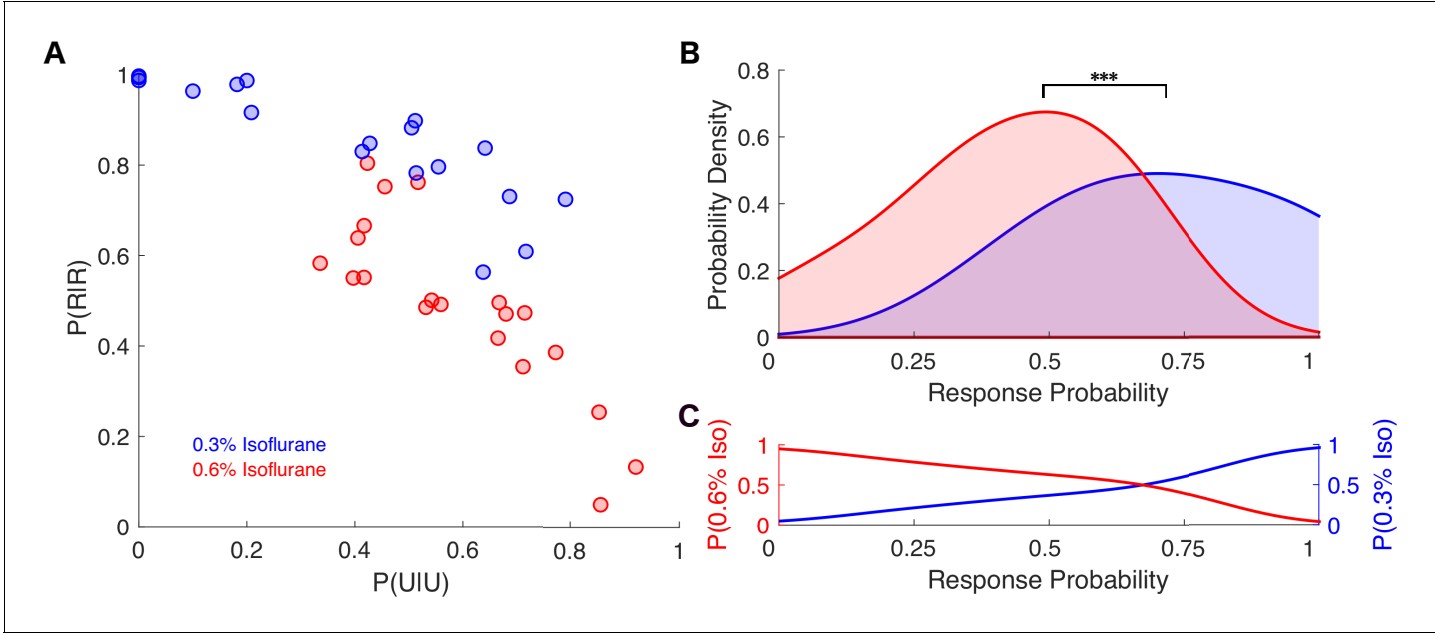

**Figure 6.** Individual response probabilities between different drug concentrations overlap at a population level. (A) Individual empirically-estimated transition probability matrices are plotted in the plane spanned by P(U|U) and P(R|R) for animals exposed to 0.3% isoflurane (n = 18, blue dots) and 0.6% isoflurane (n = 20, red dots). (B) Probability density of the individual across-trial response probabilities were estimated for mice exposed to 0.3% isoflurane (blue) and 0.6% isoflurane (red). At the population level, distributions of responsiveness at the two isoflurane concentrations are significantly different ($U = 54$, $n_{0.3\% \text{ iso}} = 18$, $n_{0.6\% \text{ iso}} = 20$, p<0.001). Overlap of the two distributions was 57%. (C) Posterior distributions, representing the probability that a mouse with a given across-trial response probability was exposed to 0.3% (blue) isoflurane or 0.6% (red) isoflurane. Over a broad range of response probabilities, the odds of correctly identifying drug concentration on the basis of observed responsiveness is close to chance. Statistical significance is shown by ***p<0.001. Source data for response probability distributions described are available in the *Figure 6—source data 1*.

The online version of this article includes the following source data and figure supplement(s) for figure 6:

**Source data 1.** PC1 values for mice exposed to 0.6% and for mice exposed to 0.3% isoflurane.
**Figure supplement 1.** Individual response probabilities between different drug concentrations overlap in zebrafish.

concentration from behavioral responses for the same individual exposed to two drug concentrations. To address this, we exposed a separate cohort of 20 mice on eight occasions, four times each to 0.4% and 0.7% isoflurane. Strong negative correlations between P(U|U) and P(R|R) existed at both 0.4% isoflurane ($R^2 = -0.69$, p<0.0001, Pearson's R) and 0.7% isoflurane ($R^2 = -0.79$, p<0.0001, Pearson's R). Population-level response probability at 0.4% and 0.7% isoflurane differed significantly (*Figure 7A*, $U = 7$, $n_{0.4\% \text{ iso}} = n_{0.7\% \text{ iso}} = 20$, p<0.0001). Overlap of response probabilities at the population level between different isoflurane concentrations did exist (18%), but was approximately three times smaller than that observed across separate mouse cohorts exposed to 0.3% and 0.6% isoflurane (*Figure 6A*). Overlap at the individual level varied widely, from close to zero to nearly complete (*Figure 7B,C*, *Figure 7—figure supplement 1*). Thus, observing the same individual exposed to different drug concentrations, improves the reliability of distinguishing between drug concentrations on the basis of righting probability. Comparing different, albeit highly genetically similar individuals, exposed to different isoflurane concentrations increases the response variability and therefore decreases the reliability of classification.

## Neuronal network modeling of stochastic state switching confirms that noise driving state transitions is conserved among individuals

In *Figures 4* and *6*, we observe a strong negative correlation between the two conditional probabilities that express the tendency of staying in the previously observed state (P(U|U) and P(R|R)). To investigate the origins of this observation, we constructed a simple mathematical model that can explain this striking correlation. The Markov process defined by a two-state transition probability matrix can be thought of as a discrete approximation of a continuous system that fluctuates

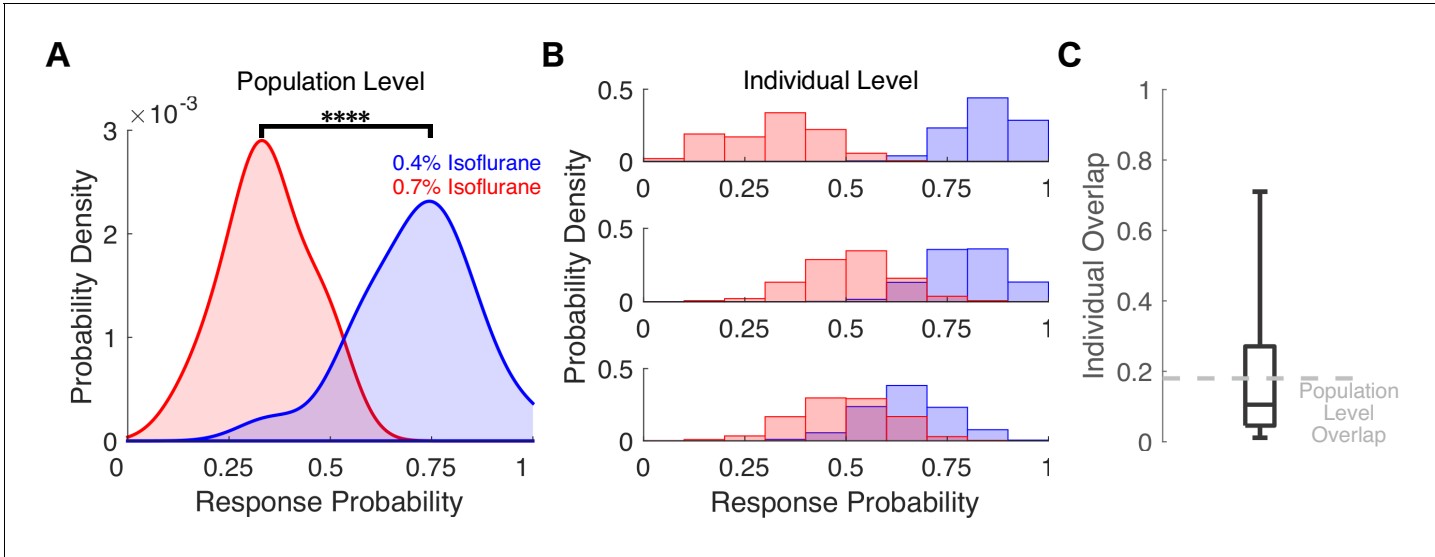

**Figure 7.** Varying degrees of overlap in response probability are present at the individual level between different isoflurane concentrations. (A) Probability density of the individual across-trial response probabilities were estimated for mice exposed to 0.4% isoflurane (blue) and 0.7% isoflurane (red). In contrast to results in Figure 6B where different populations of mice were used, here the same 20 mice were exposed to both 0.4% and 0.7% isoflurane. At the population level, distributions of responsiveness at the two isoflurane concentrations are significantly different (U = 7, n 0.4% iso = n 0.7% iso = 20, p < 0.0001). Overlap of the two distributions was 18%. (B) Probability density estimates of response probability to exposures of 0.4% (blue) and 0.7% isoflurane (red) were computed with a bootstrap resampling technique (1000 bootstraps for 20 trials with resampling). Representative probability density estimates of the same individual exposed to two drug concentrations highlighting the variability in response probability overlap across the two concentrations are shown, from marginal overlap (top), to some overlap (middle), to high overlap (bottom). (C) Boxplot representing fraction of overlap across individual mice. Box plot central mark indicates the median (11%), box shows interquartile range (5% to 27%), whiskers extend to the most extreme data points (1% and 71%). Statistical significance is shown by ****p < 0.0001. Source data for the response probability distributions described are available in the *Figure 7—source data 1*.

The online version of this article includes the following source data and figure supplement(s) for figure 7:

**Source data 1.** PC1 values for mice exposed to both 0.4% and 0.7% isoflurane.

**Figure supplement 1.** Ability to discriminate across two drug concentrations within individuals is highly variable.

stochastically between two stable states. To illustrate how such a multistable system can be embodied in the brain, we consider a simple network consisting of two neuronal populations (α and β *Figure 8A*). The two neuronal populations inhibit each other and excite themselves. *Moreno-Bote et al. (2007)* demonstrate that activity of such networks can be parameterized as a function of difference in activity of the two neuronal populations (*Figure 8B*). Because of self-excitation and mutual inhibition, the network exhibits two stable activity patterns: one where activity of α dominates, and the other, where activity of β dominates. We operationally define these stable network patterns as awake and anesthetized respectively. The likelihood of every activity pattern of the network is expressed by an 'energy function' (*Figure 8B*). The more likely activity patterns are associated with lower energy and the less likely activity patterns are associated with higher energy. In the absence of noise such networks stay in one of the two stable states indefinitely. When noise is added to the system, it switches between the two stable network configurations stochastically. While the amount of noise changes the frequency of switching between the two states (*Figure 8C*), it does not have a dramatic effect on the shape of the overall distribution of states. In both cases the distribution of states of the system is bimodal akin to what we observe in the startle reflex data in the zebrafish (*Figure 2—figure supplement 2C*).

To model the effects of anesthetics on different individuals, we assume that anesthetics activate sleep-active β neurons and inhibit wake-active α neurons. To account for the strong negative correlation between *P(U|U)* and *P(R|R)*, we assume that the degree of anesthetic-induced excitation and inhibition is correlated. Thus, individual differences in anesthetic sensitivity can be modeled by modulating the anesthetic effect on the network. In an attempt to model the results in *Figures 4* and

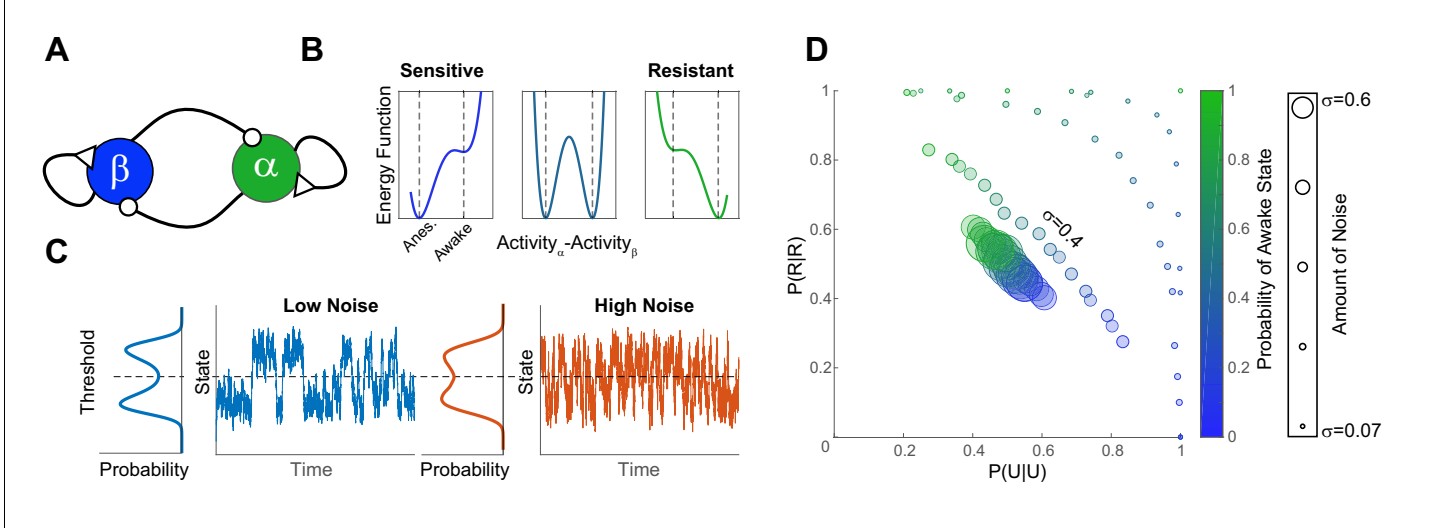

**Figure 8.** A simple bistable neuronal network model with noise, reproduces experimental findings. (**A**) The network consists of two neuronal populations (β and α) which are operationally defined as 'sleep active' and 'wake active' respectively. Neuronal populations are mutually inhibitory and self-excitatory. Activity of such networks can be parametrized as a function of difference in activity between α and β using an energy function (**B**). At population-level $EC_{50}$ (middle panel) some individuals have low probability of being in the awake state (Sensitive), while others have a high righting probability (Resistant). These differences in sensitivity are reflected in the depth of the energy wells which correspond to the Anesthetized (β-dominated) and Awake (α-dominated). The increase in the depth of the anesthetized well is proportional to the decrease in the awake well. (**C**) Shows the time evolution of network activity (at equal depths of the awake and the anesthetized wells) with different noise amounts. Trace shows the time evolution and the marginal distribution shows the probability distribution of states. Noise drives the time-evolution of the system: as more noise is added, the system fluctuates more rapidly. Since the distribution of states is bimodal, the states of the system can be discretized by applying a threshold. This is analogous to thresholding the SR responses in the zebrafish. (**D**) Discretized activity of the network was analyzed using a two-state Markov process as for experimental results. The parameters of the transition probability matrix are shown for different anesthetic sensitivities (color) and amounts of noise (symbol size). In order to obtain the data in *Figure 4C,E* and *Figure 6A* the amount of noise has to be constrained among individuals (σ = 0.4 matches the results at $EC_{50}$ for propofol and isoflurane).

*6*, we simulate the behavior of such networks with various amounts of noise. We then approximate the data of the kind seen in *Figure 8C* obtained for different anesthetic sensitivities and noise levels using a Markov model akin to that deployed for analysis of righting and startle reflex throughout this work. To accomplish this, the continuous fluctuations in the state of the network are binarized using a threshold (*Figure 8C*) into the 'responsive' and the 'unresponsive' states. The transition probability matrix is then estimated from these binary time-series. The results of this analysis are shown in *Figure 8D*. When the noise amounts are small, the system tends to stay in its previously observed state. Thus, for low-noise simulations, the points are found around the periphery of the plane spanned by *P(U|U)* and *P(R|R)*. When the noise is exceedingly high, the dynamics of the system become independent of the shape of the energy landscape and tends toward the point (0.5, 0.5) in the plot in *Figure 8C*. This would be expected if the system behaved like a Bernoulli process at $EC_{50}$. In order to reproduce the results in *Figures 4* and *6*, the amount of noise needs to be tuned and maintained at an approximately the same level across individuals. In the simulations, this corresponds to Gaussian noise with mean 0 and standard deviation (σ)=0.4. These simulations support our conjecture that while individuals are different in terms of anesthetic sensitivity, the amount of noise that drives fluctuations between the responsive and the unresponsive states is conserved. Note, that in the units of trial number, the dwell time distributions were similar in fish and in mice. However, in mice the trials were spaced 3 minutes apart whereas in fish the inter-trial interval was 30 seconds. Thus, in units of seconds, mice dwelled significantly longer in the awake and the unresponsive state than zebrafish. In order to account for this difference in the characteristic dwell times, the time scale of fluctuations shown in Figure 8C would have to be scaled by a specie-specific diffusion constant (Methods).

## Discussion

Here, we demonstrate that even when an animal is exposed to a fixed drug concentration, its response fluctuates stochastically. The probability of response depends on the state of the animal at the moment of stimulation. Specifically, the fluctuations exhibit inertia – the animal is more likely to be stuck in its current state than to transition between states of responsiveness. In contrast to most drugs that are administered either intravenously or orally, the route of administration of many anesthetics is inhalational. This allows us to assure that the drug concentration in the animal is at equilibrium with the ambient concentration in a closed chamber (*Friedman et al., 2010*). We utilize this significant experimental advantage to focus on the dynamics of responses within each individual at a constant drug concentration. However, stochastic fluctuations are not unique to a volatile anesthetic. An intravenous anesthetic, propofol, administered to zebrafish equilibrated with a fixed concentration bath, was also associated with dynamic fluctuations in responsiveness. The stochastic processes that govern the fluctuations observed under two mechanistically distinct anesthetics were remarkably similar between mice and zebrafish. It is curious to note that while the dwell times in the responsive and the unresponsive state in mice and in zebrafish were similarly correlated, the absolute duration of these states varied signficantly between mice and zebrafish. The duration of responsive and unresponsive states was approximately similar to the duration of sleep episodes in these two species (*Zhang et al., 2007*; *Yokogawa et al., 2007*). It is possible that this relationship between duration of sleep episodes and the duration of the episodes of unresponsiveness arises because similar neurobiological mechanisms are responsible for both transitions between sleep and wakefulness and fluctuations between the responsive and the unresponsive states observed under anesthesia.

Our observations of stochastic switching between responsive and unresponsive states are consistent with previous findings showing that at a fixed anesthetic concentration, spectral characteristics of electrical activity within thalamocortical networks switch stochastically between several discrete activity patterns (*Hudson et al., 2014*; *Clement et al., 2008*). Similar observations have been made using electroencephalography (EEG) of patients under anesthesia (*Chander et al., 2014*; *Li et al., 2019*; *Vlisides et al., 2019*). Our findings in this study suggest that such stochastic fluctuations have a behavioral counterpart expressed as fluctuating ability to respond to a stimulus. It has been suggested previously that anesthetics stabilize neuronal dynamics in humans (*Alonso et al., 2014*; *Tagliazucchi et al., 2016*) and in non-human primates (*Solovey et al., 2015*; *Alonso et al., 2019*). Stabilization of neuronal dynamics may contribute to the behavioral inertia observed in fluctuations in responsiveness of both mice and zebrafish. This resistance to state transitions may contribute (*Proekt and Hudson, 2018*) to anesthetic hysteresis (*Friedman et al., 2010*; *Joiner et al., 2013*; *Kuizenga et al., 2018*; *Warnaby et al., 2017*) – a left-shift in a dose-response curve for emergence relative to induction of anesthesia that has been observed across taxa from *Drosophila* to humans.

Stochastic responses at a fixed drug concentration help explain several otherwise puzzling phenomena in clinical anesthesiology. On the one hand, Hill slopes of dose-response curves for anesthetics are reported between 10 and 40 (*Miller, 2014*); some of the steepest of the clinically used drugs. This is traditionally interpreted as a sign of low inter-subject variability in responsiveness. Nonetheless, approximately ten percent of patients transiently wake up during surgery as measured by their ability to respond to a verbal command (*Sanders et al., 2017*; *Russell, 2013*; *Gaskell et al., 2017*). Luckily, incidence of awareness accompanied by post-operative recall is much less frequent (*Avidan et al., 2008*). However, even episodes of awareness associated with recall and high risk of developing post-traumatic stress disorder (*Osterman et al., 2001*) are not reliably detected by the existing EEG-based monitors of anesthetic state (*Avidan et al., 2011*). While anesthetics are thought to impart dose-dependent effects on neuronal activity (*Stanski et al., 1984*; *Katoh et al., 1998*), only weak correlations are observed between EEG characteristics and anesthetic concentration (*Whitlock et al., 2011*). Indeed, both inter-individual differences (*Whitlock et al., 2011*) and fluctuations of EEG measures of depth of anesthesia at a constant drug concentration (*Bloom et al., 2005*) obscure the relationship between drug concentration and the observed state of the EEG. It is likely that the traditional population-based approach of estimating the relationship between the concentration of the drug and the response dramatically under-represents the within and inter-subject variability. Indeed, under most circumstances, each animal is tested only once at each drug concentration. The repeated testing of each animal under a fixed drug concentration is more akin to

the clinical scenario where the patient is exposed to many surgical stimuli that can elicit stochastic fluctuations in the level of consciousness.

The Hill slope value observed in our study was approximately an order of magnitude lower than the previously published data (*Friedman et al., 2010*; *Joiner et al., 2013*). This relative decrease in Hill slope reflects high inter-individual variability and stochastic fluctuations in each individual. There are several significant differences between our experimental paradigm and those explored in the previous work. Previous work, guided by purely pharmacokinetic considerations, used shorter isoflurane exposures. In contrast, here we measured anesthetic responsiveness after a two-hour equilibration time. Under most circumstances, Markov processes lead to a single equilibrium distribution of states (*Roman, 1989*). The time it takes to reach this equilibrium, however, is dictated by, among other things, the degree of resistance to state transitions. Thus, one reason for the discrepancy between our Hill slope estimate and that published previously is that our analysis assures that the population of animals is at a behavioral steady state. In order to reliably anesthetize an individual after a short exposure, higher concentrations of anesthetics are necessary. Indeed, our $EC_{50}$ estimate (0.54–0.55% isoflurane) is significantly lower than that published in the previous work (0.9% for induction and 0.83% for emergence) (*Friedman et al., 2010*). Another fundamental difference between our approach and that used in the previous work is that we tested mice repeatedly. In contrast, in the previous work, mice are typically tested only once at each concentration. Repeated testing revealed trial-to-trial fluctuations. These fluctuations increase the apparent variability in responsiveness, thereby decreasing the apparent Hill slope of the population.

The detailed mechanism of these stochastic fluctuations is not known as anesthetics exert effects on many molecular targets distributed broadly throughout the brain and spinal cord. A number of lines of evidence converge on the fact that anesthetics at least in part hijack the sleep-wake circuitry by exciting sleep-promoting and inhibiting wake-promoting neurons (*Zhang et al., 2015*; *Moore et al., 2012*; *Vazey and Aston-Jones, 2014*; *Nelson et al., 2002*). Switching between sleep and wake is thought to be mediated by the reciprocal inhibition between these neuronal populations. Networks consisting of reciprocally inhibitory neuronal populations tend to exhibit self-reinforcing behavior. Once sleep-active neurons activate beyond a certain threshold, they shut down the wake-active neurons thereby decreasing their inhibitory effects and further strengthen mutual excitation amongst the sleep-active neurons (*Saper et al., 2005*). The converse happens during wakefulness. In the absence of perturbations, the network consisting of such mutually inhibitory self-reinforcing neuronal populations will remain in the same state indefinitely. Once sufficient noise is added, however, the system will stochastically switch between consolidated states of sleep and wakefulness. Theoretical investigations of bistable neuronal networks suggest that activity within such mutually inhibitory neuronal populations can be well approximated by a diffusion on an energy landscape with two potential wells; one for each stable activity pattern (*Moreno-Bote et al., 2007*). Anesthetics can therefore be thought of as stabilizing (deepening) the wells associated with unresponsiveness and de-stabilizing the wells associated with wakefulness. The degree to which these states are stabilized by clinically relevant doses of anesthetics is apparently insufficient to consistently keep all animals in the anesthetized state. Hence, stochastic fluctuations may be responsible for episodes of awareness that occur during surgeries.

The importance of noise in systems near a bifurcation between two stable behaviors is not limited to circuits that control sleep and wakefulness (*Chialvo, 2010*; *Destexhe and Contreras, 2006*). Neuronal architectures similar to those involved in sleep and wakefulness are thought to play a role in diverse processes such as sensory perception (*Moreno-Bote et al., 2007*), decision making (*Wong and Wang, 2006*), seizure generation (*Suffczynski et al., 2004*; *Fröhlich et al., 2010*), and working memory (*Camperi and Wang, 1998*) to name a few. Mathematically similar phenomena govern transitions between normal sinus rhythm and arrhythmias (*Kim et al., 2009*). Thus, it is likely that stochastic fluctuations between distinct responses observed under constant drug concentration are not unique to anesthetics.

To determine whether a bistable system with noise can explain the striking correlations between $P(U|U)$ and $P(R|R)$ observed herein, we simulated such a neural-network-inspired system with parametrically varied amount of noise. The results of this simulation show that in order to obtain such correlations, the amount of noise must be conserved among individuals. Another issue addressed by the modeling approach is the binary nature of behavioral responses in each individual. In zebrafish, this binarization was motivated by the observation of a bimodal distribution of distances travelled by

each zebrafish after the tap stimulus. This bimodality naturally suggests that the responses fall into two distinct classes. Yet, similar measurements are much more difficult to perform for the righting reflex. Thus, it is not clear from behavioral observations of RR alone, whether the underlying processes that give rise to a response on a RR trial are continuous or discrete. Furthermore, even if the underlying processes are discrete it is not a priori clear that only two states are present. We observed that dwell times in both the responsive and the unresponsive state are well-approximated by a single exponential distribution. Thus, we only find empirical evidence for two states. Longer recordings may be required in future work to determine whether other states are needed to completely describe the data. The two state Markov model used to analyze behavioral responses, can be seen as a natural discretization of the continuous bistable system exemplified by our modeling approach. For most reasonable choices of threshold, the continuous system and its discrete approximation will yield similar results.

In some ways, our observations of stochastic switching between different states at a fixed drug concentration are similar to those well known for single receptor molecules (*Hoshi et al., 1990*; *Papke et al., 1989*; *Hille, 2001*; *Colquhoun and Hawkes, 1995*; *Tank et al., 1982*). There is an important distinction, however, between molecular scale state transitions and those observed at the behavioral level in animals. At the molecular scale, differences between receptors are largely immaterial—the same stochastic model can be used to describe state transitions in all receptors of a particular kind. Thus, observing a single receptor molecule over time is sufficient to determine the response of a population of such receptors. Within intact multicellular organisms, in contrast, we show that the transition probabilities were significantly different amongst highly genetically similar (*Uchimura et al., 2015*) and genetically outbred individuals exposed to the same anesthetic concentration. Observation of one individual on many trials does not equate to observing a population of individuals. Indeed, at population level $EC_{50}$, some animals were three times as likely to be responsive as other individuals from the same population. Furthermore, these individual differences were consistent across time in each individual.

The inter-individual variability in the transition probabilities that govern switching between responsive and unresponsive states was constrained such that the sum of the diagonal elements was a constant. This constraint is evolutionarily conserved across vertebrates from zebrafish to mice. Note that the sum of the diagonal elements in a square matrix is known as the trace and is equal to the sum of its eigenvalues. The largest eigenvalue of a transition probability matrix for a reversible Markov process is 1 (*Levin and Peres, 2017*). The fact that the trace is approximately the same in all individuals under similar experimental conditions implies that the spectral gap of the $2 \times 2$ transition probability matrix defined as, $\lambda_1 - \lambda_2$, where $\lambda_n$ is the $n^{th}$ eigenvalue is also a constant conserved among individuals. The spectral gap of the matrix sets its mixing time, or the time it takes for a system starting out in a random distribution of states to approach its equilibrium distribution. In this particular case, the equilibrium distribution is the overall probability that a given animal will be able to respond to a stimulus. Boltzmann equation asserts that diffusive systems of the kind used in our model come to a single equilibrium distribution that depends just on the energy function. The time it takes to reach this equilibrium distribution of states, however, depends on the amount of noise. The more noise is added to the system, the quicker it reaches the equilibrium distribution. Thus, there is a fundamental relationship between the conservation of the spectral gap across individuals and the tightly controlled noise in a continuous system characterized by a two well potential. In a clear departure from the predictions made by the population-level dose-response curve, our findings indicate that the equilibrium probability of responding to a stimulus at a given drug concentration is different for distinct individuals. The time it takes to reach this behavioral equilibrium, however, is conserved amongst animals. This strongly implies that the noise which drives state transitions between responsive and unresponsive states is tightly biologically controlled. A similar observation has been made for stochastic influences on gene expression (*Elowitz et al., 2002*).

On the one hand, our results may be seen as disappointing for the clinical practice of anesthesiology. Because of the strong influence of stochastic forces, it does not appear possible to keep a subject reliably in an anesthetized state without exposing them to the potentially dangerous high concentrations of anesthetics associated with subsequently impaired cognition (*Whitlock et al., 2014*; *Chan et al., 2013*; *Fritz et al., 2016*). Yet, the fact that variability in transition probabilities is constrained offers a possible novel avenue for improvement by developing therapies specifically aimed at state stabilization. Selective stabilization of the unconscious state could provide a solution

to minimize the risk of spontaneously shifting into a brain state where awareness is possible, without requiring drug concentrations prone to adverse effects. How might this be achieved? One possibility is to target the mechanisms that are known to affect the stability of the sleep-wake circuitry. For instance, interference with orexinergic signaling destabilizes both sleep and wake states thereby increasing the frequency of spontaneous transitions between sleep and wakefulness (*Mochizuki et al., 2004*). Interestingly, interference with orexin signaling also affects responses to anesthetics (*Kelz et al., 2008*; *Shirasaka et al., 2011*; *Tose et al., 2009*; *Dong et al., 2009*). Detailed investigation of the synergy between anesthetics and modification of noise inherent in neuronal networks that control sleep and wakefulness may help develop novel therapies that will allow clinicians better control over the state of each individual patient.

## Materials and methods

### Animals

Studies were approved by the Institutional Animal Care and Use Committee at the University of Pennsylvania and were conducted in accordance with National Institutes of Health guidelines. For larval zebrafish (*Danio rerio*) experiments, Tübingen long fin wild-type zebrafish were mated (Zebrafish International Resource Center, OR), and the embryos raised for 5 days in constant darkness. At 5 days post-fertilization (dpf), the larvae were used for the experiments (n = 120). For mouse experiments, inbred male wild-type C57Bl/6 mice (Jackson Laboratories, ME) aged 16–24 weeks (n = 60) were used in righting reflex behavioral assays. One group of 20 mice was exposed to 0.6% and 0.9% isoflurane. A second set of 20 mice were exposed to 0.3% isoflurane. A third set of 20 mice were exposed to 0.4% and 0.7% isoflurane.

### Isoflurane exposure with Righting Reflex evaluation

All mice were acclimatized to sealed, temperature-controlled, 200 mL cylindrical chambers with 100% oxygen flowing at 200 mL/minute, as previously described (*Sun et al., 2006*). This flow rate ensures isoflurane chamber equilibration within 5 min (*Figure 2—figure supplement 3*). Mice were exposed to 0.90%, 0.7%, 0.60%, 0.4% or 0.30% isoflurane in 100% oxygen for 4 hr, beginning at ZT12-ZT14, corresponding to the period of maximal activity and wakefulness. Chamber isoflurane concentrations in all assays were confirmed using a Riken FI-21 refractometer (AM Bickford, NY.). As tolerance to repeated isoflurane exposures does not occur (*Smith et al., 1979*), mice were exposed to each isoflurane concentration a total of four times over the course of 3 weeks. The presence or absence of the righting reflex (RR) was checked every 3 min, starting after 2 hr of isoflurane exposure to assure pharmacokinetic equilibration. RR was assessed as described previously (*Sun et al., 2006*). A mouse was considered to have an intact righting reflex if it was able to restore its upright posture twice in a row after being turned on its back by rotating the anesthetic chamber without interrupting anesthetic delivery. Otherwise, the righting reflex was considered to be absent. Forty RR assessments were performed per animal per exposure. In total, 160 RR assessments were performed on each mouse at each anesthetic concentration.

### Propofol exposure with Startle Reflex evaluation in zebrafish

At 5 days post fertilization (dpf), individual larvae were placed into a 96-well glass plate (JG Finneran, 500 μL volume wells). Seventy-two zebrafish were exposed to 3 μM propofol in E3 medium, while 48 zebrafish were exposed to E3 alone (*Kaufman et al., 2009*). Larvae were then placed into the DanioVision (Noldus, Leesburg VA) behavioral system. Startle reflex experiments were performed at a maximal intensity tap every 30 s for 4 hr. This choice of inter-stimulus interval was based upon the lack of habituation to acoustic and vibrational stimuli in larval zebrafish when these stimuli are delivered every 15 s (*Burgess and Granato, 2007*; *Woods et al., 2014*). The histogram of the total distance travelled in the first second after each stimulus was bimodal suggesting an all-or-none response (*Figure 2—figure supplement 2C*). A threshold of 0.4 mm (approximately 1/10 of a 5 dpf zebrafish's body length), was chosen on the basis of this histogram to distinguish between responsive and unresponsive startle reflex assessments (*Kimmel et al., 1995*).

## Empirical estimation of transition probability matrices

The outcomes of the behavioral response assays, righting reflex (RR) and startle reflex (SR), are binary. The output is a series of zeros and ones (one for intact RR or SR and zero otherwise). The simplest model that describes this time series is a stochastic Markov process. Markov models relate the state of the system at time $t$ to the state of the system observed at the previous time step $t − 1$, as $X_t = MX_{t−1}$, where the 2x2 transition probability matrix, $M$, constructed as shown below is the fundamental quantity of interest.

$$M = \begin{bmatrix} a & 1-a \\ 1-b & b \end{bmatrix}$$

In the above matrix, $a$ denotes the probability that the mouse determined to be responsive on trial one will stay responsive on the next trial. $b$ denotes the probability that an animal found to be unresponsive on trial one will stay unresponsive on the next behavioral assessment. Note that because the sum of probabilities in a row of M must be 1, probabilities of transitions between the awake and the anesthetized state are completely determined by finding $a$ and $b$ in a system with just two states. The simplest possibility is that the probability of being in a particular state at time $t$ is the same for all trials (i.e. independent of the state of the system) (**Roman, 1989**). This corresponds to a scenario where, b = 1 − a. This particular type of process is called a Bernoulli process. Conversely, the future outcome of a behavioral trial may depend on the outcome of the previous trial. One manifestation of this history-dependence is inertia; the system tends to persist in a particular state. Inertia increases as the diagonal elements ($a$ and $b$) of the transition probability matrix approach one. $a$ and $b$ were empirically estimated from the binary sequence of behavioral outcomes. For instance, to calculate $a$, we computed the fraction of trials where the animal responded to the stimulus on two consecutive trials. Formally, this corresponds to P(R$_t$|R$_{t−1}$) or $P(R|R)$.

## Comparison between Bernoulli process and behavioral response

By definition, a Bernoulli process is a stochastic process in which the probability of a given outcome (positive RR or SR) is the same for every trial. This is equivalent to adapting the concept of effective concentration to an individual. For a given effective concentration, $ec$, computed as the overall probability of positive RR or SR, the Bernoulli process can be expressed as the following transition probability matrix.

$$M_b = \begin{bmatrix} ec & 1-ec \\ ec & 1-ec \end{bmatrix}$$

M$_b$ is completely specified by a single experimentally determined parameter, $ec$. For the purposes of simulations, we experimentally determined drug potency as the average response probability across all RR or SR trials and simulated the Bernoulli process given by M$_b$. M$_b$ was used to simulate 80 experiments consisting of 40 trials each to mimic experimental conditions in mice (72 experiments each consisting of 360 trials were simulated for the zebrafish experiments). Initial states were randomized in each simulated experiment. To verify that the simulated Bernoulli process and experimental observations give rise to similar drug potency, we compared the simulated and observed $ec$. To test the hypothesis that behavioral observations exhibit inertia or resistance to state transitions, we compared the observed probability of state transitions, $\frac{P(U|R)+P(R|U)}{P(R|R)+P(U|U)+P(U|R)+P(R|U)}$, in the observed and the simulated time series. As an additional test of adequacy for describing experimentally observed time series by a Bernoulli process, we compared the tendency of the system to stay in the unresponsive state, $P(U|U)$, to probability of becoming unresponsive after being responsive on the previous trial, P(U|R).

## Inter-individual variability

The transition probability matrix for a two-state Markov process has two free parameters. A transition probability matrix estimated for each specific individual can therefore be represented by a point on a plane spanned by the diagonal elements of the matrix. Principal component analysis (PCA) was used to capture the maximal inter-individual variance between transition probability matrices. To determine whether the transition probability matrix is a consistent trait of each individual animal, we

determined the correlation in position of each individual along the first principal component (PC) across time. Statistical significance of this correlation was assessed using a permutation test (10,000 permutations).

## Decoding drug concentration from individual responsiveness

To quantify how reliably the drug concentration can be inferred from the observed behavioral responses of each individual, we applied Bayes' theorem:

$$P(Drug|Response) = \frac{P(Response|Drug)\,P(Drug)}{P(Response)}$$

This analysis was performed to determine how reliably one can distinguish between exposures to 0.6% and 0.3% isoflurane in mice or 0 vs. 3 μM propofol in zebrafish.

Since mice exposed to 0.6% and 0.3% isoflurane were from two separate populations, we wanted to further investigate whether it is possible to infer drug concentration from behavioral responses within an individual exposed to two drug concentrations. For this purpose, we exposed a separate cohort of twenty mice on eight occasions, four times each to 0.4% and 0.7% isoflurane. Here, the collective trials at each concentration were pooled together for each individual (160 trials), and a bootstrap resampling technique with replacement was used to generate a response probability distribution for each concentration for each individual mouse. 1000 bootstraps consisting of 20 randomly chosen trials were taken, where the average response probability of each bootstrap was recorded. Probability distribution functions were then fit to each mouse's bootstrapped response probabilities for both isoflurane concentrations. Overlap between the two concentrations was computed for each mouse by finding the union of the two distributions. At the population level, response probability distributions were computed based on average response probabilities for each animal, and overlap was computed in a similar fashion.

## Hill slope estimation from steady state population response probabilities

The population response probabilities observed in *Figure 2B* and *Figure 2—figure supplement 1* suggest a shallower dose-response as compared to previously published population-based dose-response curves (*Friedman et al., 2010*; *Joiner et al., 2013*; *Sun et al., 2006*). In order to compute the aggregate dose-response, we fit sigmoid curves to jackknifed subsamples across all tested isoflurane concentrations. Data from one animal at each concentration was removed for each subsample, and the remaining subset of data was fitted to a sigmoidal curve:

$$1 - R = \frac{1}{1 + (EC_{50}/Iso)^H}$$

where R is the righting probability, Iso is the isoflurane concentration and the two paramters $EC_{50}$ and H are the half-maximal effect concentration and the Hill slope respectively. Fits were calculated such that the sum of the squared error was minimized between the subsample values and sigmoid. We constrained the fit such that the sigmoid value was 1 at 0.0% isoflurane, and 0 at 0.9% isoflurane. Mean and 95% confidence intervals were calculated for the $EC_{50}$ and the Hill slope.

## Modeling of a bistable neuronal network

The modeling approach here is essentially identical to that in *Proekt and Hudson (2018)* and *Moreno-Bote et al. (2007)*. Activity patterns of the bistable neuronal network consisting of two neurons can be parametrized as a function of the difference between activity of α neurons and β neurons $x = Activity_\alpha\text{-}Activity_\beta$. To express the fact that mutual inhibition and self-excitation give rise to the winner-take-all behavior, *Moreno-Bote et al. (2007)* used the following energy function that we adapt here:

$$E(x) = x^2\left(\frac{X^2}{2} - 2\right)$$

This function has two energy minima x=(−1, 1) corresponding to β-dominated ('anesthetized')

and α-dominated ('awake') activity states, respectively. To add the effects of anesthetics we modified this function as follows (*Proekt and Hudson, 2018*)

$$E(x,i) = x^2\left(\frac{X^2}{2} - 2\right) + i(x-1)^2 + (1-i)(x+1)^2$$

where $i$ is the anesthetic concentration i $\in$ [0, 1] in arbitrary units. Note that in this equation, the effect of anesthetic on stabilizing the anesthetized state is proportional to that destabilizing the awake state. This was done to reflect the results shown in *Figures 4* and *6* which show strong negative correlation between $P(U|U)$ and $P(R|R)$ in both mice and zebrafish. Individual differences in righting probability were simulated by changing $i$. This results in deepening the energy well in the vicinity of $x=-1$ and making the well in the vicinity of $x = 1$ proportionally shallower. Time evolution of the state of the network, $x$, was simulated as diffusion on an energy function using the standard approach:

$$\frac{dx}{dt} = -D\frac{\partial E(x,i)}{\partial x} + \epsilon$$

where $D$ is the diffusion constant (assumed to be one in all simulations for simplicity). The first term in the equation reflects the energy gradient. This assures that the system tends towards energy minima. The second term, $\epsilon$, is zero mean Gaussian noise. To modulate the amount of noise in the system, the standard deviation, $\sigma$, of noise was altered parametrically. Simulations of this diffusion equation were performed for 1,000,000 time steps. While $x$ varies continuously, the distribution of $x$ is bimodal with peaks near the energy troughs $x=(-1, 1)$. To compare simulations to our experimental results on RR and SR modeled using a two-state Markov process, we binarized $x$ such that $x > 0$ was classified as 'anesthetized' and $x \leq 0$ was classified as 'awake'. To further mimic the fact that RR and SR are performed intermittently, the binarized time series from the simulation was decimated 100 fold. This binary time series was modeled by a two-state transition probability matrix using the same methods as for the analysis of the RR and SR. The results of these simulations for different anesthetic concentrations and $\sigma$'s is shown in *Figure 8D*.

## Statistical analyses

Analyses were performed using custom code written in Matlab using the Statistics and Machine Learning toolboxes (Mathworks, MA). Steady state population response probabilities were confirmed through Pearson correlation coefficient analysis between the trial index and the response probability averaged across animals. $R^2$ values approaching zero indicate that the population is near a steady state. Statistical comparisons of medians were performed using the nonparametric Mann-Whitney U test. The Brown-Forsythe test for equal variances was used to compare variability in simulated and experimental response probabilities. $p < 0.05$ were considered statistically significant for all comparisons.

## Acknowledgements

We are grateful to Andrew Hudson, Connor Brennan, Sarah Reitz, Adeeti Aggarwal, and Brenna Shortal for their valuable discussion and suggestions about the data.

This work was supported by the National Institute of Health grants K08 GM123317, R01 GM124023, R01 GM088156, R01 GM107117, T32 GM112596, and by the Department of Anesthesiology and Critical Care at the University of Pennsylvania.

## Additional information

### Funding

| Funder | Grant reference number | Author |
| --- | --- | --- |
| National Institutes of Health | K08 GM123317 | Andrew R McKinstry-Wu |
| National Institutes of Health | R01 GM124023 | Alexander Proekt |

| National Institutes of Health | R01 GM088156 | Max B Kelz |
| National Institutes of Health | R01 GM107117 | Max B Kelz |
| National Institutes of Health | T32 GM112596 | Max B Kelz |

The funders had no role in study design, data collection and interpretation, or the decision to submit the work for publication.

## Author contributions

Andrew R McKinstry-Wu, Conceptualization, Resources, Supervision, Funding acquisition, Methodology, Writing—original draft, Writing—review and editing; Andrzej Z Wasilczuk, Conceptualization, Data curation, Software, Formal analysis, Validation, Investigation, Visualization, Methodology, Writing—original draft, Writing—review and editing; Benjamin A Harrison, Data curation, Validation, Investigation, Methodology; Victoria M Bedell, Resources, Data curation, Investigation, Methodology, Writing—review and editing; Mathangi J Sridharan, Data curation, Investigation, Methodology; Jayce J Breig, Investigation, Methodology; Michael Pack, Resources, Supervision, Methodology; Max B Kelz, Conceptualization, Resources, Supervision, Funding acquisition, Methodology, Writing—review and editing; Alexander Proekt, Conceptualization, Resources, Software, Formal analysis, Supervision, Funding acquisition, Validation, Methodology, Writing—original draft, Writing—review and editing

## Author ORCIDs

Andrzej Z Wasilczuk https://orcid.org/0000-0002-1492-2319
Alexander Proekt https://orcid.org/0000-0002-9272-5337

## Ethics

Animal experimentation: Studies were approved by the Institutional Animal Care and Use Committee protocol (803844) at the University of Pennsylvania and were conducted in accordance with National Institutes of Health guidelines.

## Decision letter and Author response

Decision letter https://doi.org/10.7554/eLife.50143.sa1
Author response https://doi.org/10.7554/eLife.50143.sa2

# Additional files

## Supplementary files

• Source code 1. Estimation of Dwell Time Distributions.

• Source code 2. Estimation of Transition Probability Matrix.

• Source code 3. Simulation of a Markov Process.

• Transparent reporting form

## Data availability

All data generated or analysed during this study are included in the manuscript and supporting files. Raw data and source data files have been provided as Supplementary files.

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
