## [Decision Letter]

**Acceptance summary:**

This study examines intra- and inter-subject variability in anaesthetic efficacy in two animal species, mice and zebrafish. The authors find that in the vicinity of the EC_50_ for two different anaesthetics, animals exhibit state-dependent fluctuations between putative conscious and unconscious states. Importantly, the analysis separates trial- and individual-level variability in behavioral outcomes, showing that population level measurements of drug efficacy can be very unreliable measures of efficacy at the individual level, and that individual fluctuations are consistent. This has important implications for personalised medicine and anaesthesia in practice.

**Decision letter after peer review:**

Thank you for submitting your article "Tight control of noise in state transitions revealed by dynamics of fluctuating individual drug responses" for consideration by *eLife*. Your article has been reviewed by three peer reviewers, one of whom is a member of our Board of Reviewing Editors, and the evaluation has been overseen by Richard Aldrich as the Senior Editor. The following individual involved in review of your submission has agreed to reveal their identity: Robert Pearce (Reviewer #3).

The reviewers have discussed the reviews with one another and the Reviewing Editor has drafted this decision to help you prepare a revised submission.

All reviewers agreed that the work is important and that the manuscript should appeal to a large cross section of life scientists. The reviews highlighted a few essential revisions (in detail below) that will improve the manuscript and facilitate readers. These concern the interpretation of the results, the assumptions, statistical analyses and a few points on presentation.

Reviewer #1:

This study examines intra- and inter-subject variability in anaesthetic efficacy in two animal species, mice and zebrafish. The authors find that in the vicinity of the EC_50_ for two different anaesthetics, animals exhibit state-dependent fluctuations between putative conscious and unconscious states. The experiments seem to be conducted rigorously and the manuscript is clear.

It is not clear to me that the pharmacological community (or life scientists and clinicians more generally) would be surprised by the results of this study. Nonetheless, there is a temptation to summarise population effects without considering individual variability or time (state dependence). I think the study is valuable because it illustrates both. With minor editing it could be clearer still, and will help members of the community avoid pitfalls of interpreting dose-response relationships naively.

I have only one substantive concern that is easily addressed. The statistical analyses are not reported adequately. It is not sufficient to simply report a p-value; the degrees of freedom in the test should be provided to ensure appropriate units of replication.

Reviewer #2:

McKinstry-Wu et al. report a model of behavior during anesthesia aiming to account for variability within each behavioral state. This paper presents a useful examination of the variable and stochastic nature of behavior during anesthesia, separating trial- and individual-level variability in behavioral outcomes. A concern, which could be addressed with rewriting, is that the authors discuss anesthesia as if it were intrinsically an all-or-none brain state, but their choice of experimental design imposes the binomial structure onto the data.

Results:

- The use of within-session comparisons for the zebrafish condition (Figure 5), as opposed to the cross-session testing in mice, holds the limitation that there will be session-dependent effects included in this analysis.

- For Figure 6, do the same mice exhibit high sensitivity in both conditions? i.e., within an individual does this overlap still hold? This overlap analysis is interesting but it is not clear from the figure whether it is due to variability within the population or within a given concentration.

Comments on the text:

- The paper's Introduction states that anesthesia is a binary state, and that an individual is either anesthetized or not. However, the state of anesthesia is a spectrum with multiple dimensions – for example, a patient can be in a lightly anesthetized state in which they will not react to a simple sensory stimulus, but will still react to a painful stimulus. The Introduction should be written to clarify this distinction. The method the authors propose can be applied to a specific desired behavioral outcome of interest during anesthesia, but it doesn't mean that anesthesia itself is binary.

- While the authors state several times that behavior is thought to be constant during anesthesia, in general it is known that behavior is variable at a given concentration, for example a binomial state-space model of variable behavior during multiple concentrations of anesthesia has been previously developed and should be cited, Wong et al., 2014.

- The discussion of bistable states is a bit misleading, particularly the section comparing sleep and anesthesia, since it seems unlikely this is a bistable switching between sleep and wakefulness. Even a very gradual descent into anesthesia will appear bistable if a single yes/no threshold is applied to define the behavioral state. In this scenario, the bistable nature is imposed by the behavioral reporting scheme used, and a completely different result could be obtained by using a different behavioral paradigm (for example, if using a study measuring reaction time or magnitude of response). The methods point out the thresholding techniques applied to achieve a binomial outcome (a decision as to whether the mouse achieves 2 separate rightings, and a threshold of distance travelled that is applied to the zebrafish). The Figure 2—figure supplement 2 provides an example of this, as while there is bimodality to the behavior, the yes/no behavioral report is imposed by the authors' threshold. It's useful to distinguish between the bistability of the behavioral test (which is useful for clinical purposes) vs. the bistability of the brain state (which is not shown here, since no neural data were acquired, and in general is found to exhibit complex graded dynamics across anesthetic states).

Reviewer #3:

This paper describes experiments that characterize responsiveness of mice (righting reflex) and zebrafish (startle reflex) in the presence of varying concentrations of anesthetics. The investigators find that i) responsiveness does not match expectations of a Bernoulli process, but rather it fluctuates on a time scale slower than their repeated observations; ii) anesthetic sensitivity differs between individuals, both on a short-term and on a long-term basis; and iii) response probability shows a broad distribution for individuals, in contrast to the expected steep dose-response of population responses. Based on similarities between inter-individual transition variability characteristics the authors argue that noise driving transitions is a conserved and tightly constrained value across species.

The study has many strengths. It uses an interestingly different approach to address an old question, one that has been investigated extensively in the past but that remains unanswered and of intense interest. The methodology is sound, and the observations have been carefully and thoroughly documented. The analysis based on probabilistic statistics makes a convincing case for fluctuating responsiveness that differs between individuals, a finding that is novel and important to the field.

One aspect of the paper that I find less convincing, and possibly problematic, is that the Abstract and Discussion focus so strongly on the 'noise' driving transitions, concluding that noise is tightly controlled and conserved across species – all without having made any explicit assessment of the inferred noise nor any attempt to manipulate it. Rather, the conclusion was based on the finding that the joint distributions of transition probabilities lie on a diagonal and are strongly negatively correlated (Figure 4). The interpretation that 'noise' drives the transitions depends on the underlying model. The authors do put forward a possible model, with their suggestion that the consolidated states of 'sleep' and 'wakefulness' may form such a neurophysiological model. In their model, noise is required to overcome a barrier between the states, similar to the barrier between metastable states of a receptor. This is a reasonable, even attractive, model, but it is conceptual rather than quantitative. Thus, it is difficult to understand how this might be applied to state changes in different individuals or species, and whether the amount of noise is the same or not. Perhaps a quantitative model of the depths of wells/heights of barriers could be derived from the data, using the data of Figure 4—figure supplement 1 to derive transition rates, and compare them explicitly between individuals and species.

Another interesting observation that is discussed relatively extensively is the relationship between individual responsiveness, which seems to show a relatively shallow concentration-response, compared to the expected steepness based on previous population data. It would be useful to know whether the aggregated data do indeed show the expected (as quoted) Hill coefficient of 10-40. If this is the case, can the authors explain how dispersed shallow concentration responses add up to steep population response relationships? If it is not the case, this information is worth noting and it will further inform the discussion.

---

## [Author Response]

Reviewer #1:[…] It is not clear to me that the pharmacological community (or life scientists and clinicians more generally) would be surprised by the results of this study. Nonetheless, there is a temptation to summarise population effects without considering individual variability or time (state dependence). I think the study is valuable because it illustrates both. With minor editing it could be clearer still, and will help members of the community avoid pitfalls of interpreting dose-response relationships naively.I have only one substantive concern that is easily addressed. The statistical analyses are not reported adequately. It is not sufficient to simply report a p-value; the degrees of freedom in the test should be provided to ensure appropriate units of replication.

We have added appropriate statistical measures for all comparisons.

Reviewer #2:McKinstry-Wu et al. report a model of behavior during anesthesia aiming to account for variability within each behavioral state. This paper presents a useful examination of the variable and stochastic nature of behavior during anesthesia, separating trial- and individual-level variability in behavioral outcomes. A concern, which could be addressed with rewriting, is that the authors discuss anesthesia as if it were intrinsically an all-or-none brain state, but their choice of experimental design imposes the binomial structure onto the data.Results:- The use of within-session comparisons for the zebrafish condition (Figure 5), as opposed to the cross-session testing in mice, holds the limitation that there will be session-dependent effects included in this analysis.

It is indeed possible that there are session specific effects in zebrafish. As we directly state in the manuscript, developmental confounds prevent us from studying the same zebrafish at widely separated time points.

- For Figure 6, do the same mice exhibit high sensitivity in both conditions? i.e., within an individual does this overlap still hold? This overlap analysis is interesting but it is not clear from the figure whether it is due to variability within the population or within a given concentration.

This is an excellent point. Thank you. To address it we had to perform additional experiments as in the original manuscript, two different cohorts of mice were subjected to 0.3 and 0.6% isoflurane. We now subjected a separate cohort of mice to both 0.4 and 0.7% isoflurane and report the results in the revised manuscript. When the mice are compared to themselves at different isoflurane concentrations the overlap (shown in newly added Figure 7) is still present. However, it is smaller than that in Figure 6. Using bootstrapping we estimated the response distributions for each individual and compared the distributions observed at 0.4 and 0.7% isoflurane on an individual basis. The results revealed variable degree of overlap at the level of an individual. The comparatively smaller overlap in this new experimental group is consistent with our observations that specific parameters of fluctuations are a unique identifiable feature of an individual.

Comments on the text:- The paper's Introduction states that anesthesia is a binary state, and that an individual is either anesthetized or not. However, the state of anesthesia is a spectrum with multiple dimensions – for example, a patient can be in a lightly anesthetized state in which they will not react to a simple sensory stimulus, but will still react to a painful stimulus. The Introduction should be written to clarify this distinction. The method the authors propose can be applied to a specific desired behavioral outcome of interest during anesthesia, but it doesn't mean that anesthesia itself is binary.

We agree that, in part the binary nature of the responses is forced upon the data by our measurement of the righting (or startle) reflex. The reviewer is correct to say that the state of anesthesia is a “spectrum with multiple dimensions”. We now discuss this important caveat in the Introduction to the manuscript in detail. That being said, while it is true that responses to anesthetics can be thought of as a spectrum, several lines of evidence suggest that our binarized definition of the anesthetic state is not without merit.

First, the most universally used measure of anesthetic potency is minimal alveolar concentration (MAC). MAC was originally defined as the concentration at which 50% of subjects lose their ability to respond to painful surgical stimuli. Subsequently other MAC-values have been defined. For instance, MAC-awake defines the concentration at which 50% of human subjects lose their ability to respond to verbal commands. This MAC-awake value is very closely correlated with MAC for righting reflex in mice for a number of mechanistically distinct anesthetics Franks 2008. Thus, while it is true that multiple anesthetic end points can be defined, with respect to the most common measures used for several decades in anesthesia research, the responses of each individual are binary.

The second reason to suspect that the state of general anesthesia may not be entirely graded is that during exposure to a fixed anesthetic concentration, brain activity stochastically fluctuates between several discrete states Hudson et al., 2014. The discrete nature of fluctuations in brain activity under anesthesia has been confirmed with cluster analysis. Similar fluctuations among discrete activity patterns are observed in human EEG during surgery Chander et al., 2014. Here we show that, much like brain activity, behavioral responses also fluctuate at a fixed drug concentration.

While it is true that one could devise a continuous measure of behavioral responses such as response latency suggested by the reviewer, this measurement is not routinely performed in a clinical setting. Finally, the ultimate goal of administering anesthesia is to render the patient unconscious. The increase in the reaction time that would most likely be observed with low doses of anesthetics does not distinguish between conscious and unconscious patient. Patients capable of responding to the stimulus, albeit with longer latency, are nevertheless awake by definition. While little is known about the neuronal correlates of consciousness, it is nevertheless reasonable to assume that there is a discrete difference between the conscious and the unconscious state of the brain.

We have added a continuous bistable system into the manuscript. This model illustrates the relationship between the discrete responses defined on the basis of behavior and the underlying continuous dynamics (see details below).

- While the authors state several times that behavior is thought to be constant during anesthesia, in general it is known that behavior is variable at a given concentration, for example a binomial state-space model of variable behavior during multiple concentrations of anesthesia has been previously developed and should be cited, Wong et al., 2014.

Thank you. We have added this reference to this manuscript in the Introduction.

- The discussion of bistable states is a bit misleading, particularly the section comparing sleep and anesthesia, since it seems unlikely this is a bistable switching between sleep and wakefulness. Even a very gradual descent into anesthesia will appear bistable if a single yes/no threshold is applied to define the behavioral state. In this scenario, the bistable nature is imposed by the behavioral reporting scheme used, and a completely different result could be obtained by using a different behavioral paradigm (for example, if using a study measuring reaction time or magnitude of response). The methods point out the thresholding techniques applied to achieve a binomial outcome (a decision as to whether the mouse achieves 2 separate rightings, and a threshold of distance travelled that is applied to the zebrafish). The Figure 2—figure supplement 2 provides an example of this, as while there is bimodality to the behavior, the yes/no behavioral report is imposed by the authors' threshold. It's useful to distinguish between the bistability of the behavioral test (which is useful for clinical purposes) vs. the bistability of the brain state (which is not shown here, since no neural data were acquired, and in general is found to exhibit complex graded dynamics across anesthetic states).

We agree that thresholding of the behavioral data is a non-trivial issue. Also, please see our response to Major Comment 3 with respect to the discrete nature of brain activity under anesthesia. We acknowledge the thresholding issue in the Discussion section. We also address the issue of thresholding in a mathematical model (see responses to reviewer 3) newly added to the manuscript. This model does not a prioriassume the discreteness of responses and explicitly introduces a threshold for binarizing behavior. Our choice to binarize the data is a practical one. As noted by the reviewer, while movement of the zebrafish in response to a tap stimulus is continuous, the distribution of the net amount of movement is clearly bimodal. The implication of this bimodality is that movements elicited by the startle stimulus fall into two distinct classes: responsive and unresponsive. Our goal here was not to precisely model the kinematics of each specific response but to more coarsely describe the system as fluctuating between the responsive and the unresponsive state. No experimental observation is purely binary. Even in the presence of perfectly binarized responses, measurement error will necessarily contaminate the results and yield a bimodal distribution, which is what we observe in the zebrafish. This for instance is also true for single ion channel recordings: the histogram of conductance levels is multimodal.

We use the following observations to assert that stochastic fluctuations between two states is a parsimonious model that can account for the data:

1) Dwell times in each state are exponentially distributed.

2) No autocorrelations in fluctuations are observed.

3) Deviations from a Bernoulli process imply stabilization of the unresponsive state.

4) A continuous bistable model when subjected to a threshold (akin to that used for zebrafish data) recapitulates all experimental results (including the correlation between P(U|U) and P(R|R).

It is possible that a more complex model will be required in the future. However, all of the present data are readily explained by noise driven fluctuations of a bistable system.

Reviewer #3:[…] One aspect of the paper that I find less convincing, and possibly problematic, is that the Abstract and Discussion focus so strongly on the 'noise' driving transitions, concluding that noise is tightly controlled and conserved across species – all without having made any explicit assessment of the inferred noise nor any attempt to manipulate it. Rather, the conclusion was based on the finding that the joint distributions of transition probabilities lie on a diagonal and are strongly negatively correlated (Figure 4). The interpretation that 'noise' drives the transitions depends on the underlying model. The authors do put forward a possible model, with their suggestion that the consolidated states of 'sleep' and 'wakefulness' may form such a neurophysiological model. In their model, noise is required to overcome a barrier between the states, similar to the barrier between metastable states of a receptor. This is a reasonable, even attractive, model, but it is conceptual rather than quantitative. Thus, it is difficult to understand how this might be applied to state changes in different individuals or species, and whether the amount of noise is the same or not. Perhaps a quantitative model of the depths of wells/heights of barriers could be derived from the data, using the data of Figure 4—figure supplement 1 to derive transition rates, and compare them explicitly between individuals and species.

We absolutely agree with the reviewer that noise ended up playing an outsized role in how the data were presented in the original manuscript. We de-emphasized the significance of noise. We also agree that quantification and a more formal definition of noise was lacking. To address this shortcoming, we introduced a mathematical model that summarizes our findings and defines noise more rigorously.

We chose to describe the system as stochastic switching between two states: responsive and unresponsive. This choice is motivated by the observation that the dwell time distribution (both mice and fish) in the responsive and the unresponsive states were approximately exponential and no autocorrelations in fluctuations were found. Thus, stochastic state transitions are the simplest model that can account for the data.

Specifically, the Markov process that we used in the paper can be thought of as a discrete approximation to a continuous system with noise. In order to account for the apparent resistance to state transitions (see Figure 3), a continuous model must have a stabilization mechanism. One of the simplest examples of such systems with relevance to neuroscience is a system of mutually inhibitory neuronal populations with self-excitation Moreno-Bote et al., 2007. Theoretical work on such winner-take all networks suggests that the state space of all activity patterns exhibited by them can be characterized by an energy landscape (see newly added Figure 8). Each well in the energy landscape corresponds to an activity pattern where one of the neuronal groups is active and the other is silent. The dynamics of such networks can then be modeled as a diffusive process on this energy landscape. Because in our work we found evidence for just two states (each dwell time is approximately exponential) we modeled the system using two mutually inhibitory “wake active” and “sleep active” neuronal populations. Effects of anesthetics can be modeled as exciting “sleep active” neurons and inhibiting “wake active” ones. Because we observe that the individual differences in anesthetic responses in both mice and in zebrafish are constrained such that stability of the anesthetized state (P(U|U)) is negatively correlated with stability of the awake state (P(R|R)), we modeled anesthetic-induced excitation of sleep active and inhibition of wake active neurons as proportional.

Using this model, we can now examine the effect of noise. This was performed by simulating diffusion on an energy landscape using a stochastic differential equation where the gradient of the energy landscape and (Gaussian) noise drive the time-evolution of the state of the network. Newly introduced Figure 8 shows the results of this analysis. As expected, when the noise is low the system tends to dwell in its previously observed state. If noise is high, the system switches state rapidly. While the state of the system is continuous, the distribution of states is multimodal (Figure 8C). This is in precise accordance to what we observe in zebrafish. This also addresses the concerns of reviewer 2 regarding binarization of continuous data. By finding the gap between the two modes we threshold the data (exactly as in zebrafish experiments) and approximate the resultant binarized system using a two-state Markov process in the same way as in the zebrafish and mouse behaviors. The results in Figure 8D are in precise accordance with the observed data. When the noise is low, the systems characterized by different drug sensitivities are found around the outer periphery of the plane spanned by P(U|U) and P(R|R). When the noise is high, the dynamics become dominated by noise and the systems tend to the point (0.5 0.5) in this plot. Only when noise is constrained to an intermediate level, do we observe a line of points that characterizes the individual differences between mice and zebrafish. This is the foundation for our claim that, in order to observe the experimental data (Figure 4 and 6) the noise that drives fluctuations between states of responsiveness must be constrained. It is possible that a more sophisticated model could also reproduce the results, yet this simple model is sufficient and therefore we adapt it in our interpretation of the data.

As you suggested, we studied the dwell time distributions in both mice and in fish by fitting them to exponential functions. We observe that much like in the case of the transition probabilities (P(U|U) and P(R|R) the time constant for the dwells in the responsive and the unresponsive states are strongly negatively correlated. We now added this important observation into the manuscript. More generally, the specifics of the dwell time distribution will depend on the shape of the energy function and not just the depths of the wells. Since we do not have any means to measure the shape of the energy wells, we did not seek to formally relate the time constants for dwell time distribution to the height of the energy barrier. That being said, we found that standard deviation of 0.4, when added to the bistable system most closely approximated the results in both mice and in zebrafish at ~ EC_50_ (Figure 8D). We also compared the sum of the decay rates for the responsive and the unresponsive states in mice and in zebrafish. This sum of decay rates is a proxy for the depths of the energy wells. We failed to detect any significant differences between fish and mice. This further confirms the intuition that the amount noise, relative to the height of the energy barrier, is conserved.

Another interesting observation that is discussed relatively extensively is the relationship between individual responsiveness, which seems to show a relatively shallow concentration-response, compared to the expected steepness based on previous population data. It would be useful to know whether the aggregated data do indeed show the expected (as quoted) Hill coefficient of 10-40. If this is the case, can the authors explain how dispersed shallow concentration responses add up to steep population response relationships? If it is not the case, this information is worth noting and it will further inform the discussion.

Like you, we were also intrigued by the stark difference between our estimates of inter subject variability and the steep population dose-response curve that has been noted in previous studies. To answer your question, we performed additional experiments on a cohort of 20 mice that were subjected (in the same way as in the rest of the manuscript) to 0.4 and 0.7% isoflurane. Combining the data from 0.3, 0.4, 0.6, 0.7, and 0.9% isoflurane concentrations and modeling the results using the standard Hill equation yields a slope of ~2. This is consistent with our observations of inter-subject variability but is clearly lower than the previously published data. It is presently unclear how this difference arises and how to reconcile the previously published results with the ones presented herein. One of the fundamental differences is that we test the righting reflex repeatedly. Repeated testing yields fluctuating results which contribute to the variability of the response. Because in previous studies the response was typically tested only once per anesthetic concentration, the variability appears much smaller. Also, as we now discuss in the manuscript, the fluctuations in responsiveness introduce time-dependence. This time dependence reflects the rate at which a stochastic process approaches its equilibrium distribution. Long exposures used in our study yield a behavioral equilibrium (for a population of mice). Short exposures used previously may not. This is consistent with our finding that the EC_50_ for RR is considerably less in our work than in previously published studies. We now discuss this in the manuscript directly.